# Fun30 and Rtt109 Mediate Epigenetic Regulation of the DNA Damage Response Pathway in *C. albicans*

**DOI:** 10.3390/jof8060559

**Published:** 2022-05-25

**Authors:** Prashant Kumar Maurya, Pramita Garai, Kaveri Goel, Himanshu Bhatt, Anindita Dutta, Aarti Goyal, Sakshi Dewasthale, Meghna Gupta, Dominic Thangminlen Haokip, Sanju Barik, Rohini Muthuswami

**Affiliations:** 1Chromatin Remodelling Laboratory, School of Life Sciences, Jawaharlal Nehru University, New Delhi 110067, India; prashantmaurya07@gmail.com (P.K.M.); garai.pramita@gmail.com (P.G.); kaverigoel.29@gmail.com (K.G.); himanshu.chandra.dutt.bhatt@gmail.com (H.B.); aninditasls1991@gmail.com (A.D.); aartigoyal96@gmail.com (A.G.); sakshidewasthale@gmail.com (S.D.); sanjubarik24@gmail.com (S.B.); 2Department of Biochemistry and Biophysics, University of California, San Francisco, CA 94158, USA; gupta@msg.ucsf.edu; 3Department of Botany, United College, Chandel, Manipur University, Mizoram 795172, India; domlenz62@gmail.com

**Keywords:** Rtt109, Fun30, chromatin remodeling, DNA damage response, *C. albicans*

## Abstract

Fun30, an ATP-dependent chromatin remodeler from *S. cerevisiae*, is known to mediate both regulation of gene expression as well as DNA damage response/repair. The Fun30 from *C. albicans* has not yet been elucidated. We show that *C. albicans* Fun30 is functionally homologous to both *S. cerevisiae* Fun30 and human SMARCAD1. Further, *C. albicans* Fun30 can mediate double-strand break end resection as well as regulate gene expression. This protein regulates transcription of *RTT109*, *TEL1*, *MEC1*, and *SNF2*-genes that encode for proteins involved in DNA damage response and repair pathways. The regulation mediated by *C. albicans* Fun30 is dependent on its ATPase activity. The expression of *FUN30*, in turn, is regulated by histone H3K56 acetylation catalyzed by Rtt109 and encoded by *RTT109*. The *RTT109Hz*/*FUN30Hz* mutant strain shows sensitivity to oxidative stress and resistance to MMS as compared to the wild-type strain. Quantitative PCR showed that the sensitivity to oxidative stress results from downregulation of *MEC1*, *RAD9*, *MRC1*, and *RAD5* expression; ChIP experiments showed that Fun30 but not H3K56ac regulates the expression of these genes in response to oxidative stress. In contrast, upon treatment with MMS, the expression of *RAD9* is upregulated, which is modulated by both Fun30 and H3K56 acetylation. Thus, Fun30 and H3K56 acetylation mediate the response to genotoxic agents in *C. albicans* by regulating the expression of DNA damage response and repair pathway genes.

## 1. Introduction

The ATP-dependent chromatin remodeling proteins modulate gene expression as well as mediate DNA damage response/repair by using the energy released from ATP hydrolysis [1]. Though classified as helicases and possessing the seven conserved helicase motifs, these proteins do not possess canonical helicase activity [2,3]. Instead, they use the energy stored in ATP to reposition, evict, or slide nucleosomes [4]. In addition, they can also mediate histone variant exchange [5].

The ATP-dependent chromatin remodeling proteins are classified into 24 subfamilies, of which the Etl1 subfamily comprises Etl1 in mice, SMARCAD1 in humans, Fun30 in *S.*
*cerevisiae*, and Fft1, Fft2, Fft3 in *S. pombe* [2]. All the members of the subfamily have been shown to mediate DNA end resection during double-strand break repair, thus indicating that these proteins are involved in the repair process [6,7]. Studies have also shown that these proteins are required for maintaining the chromatin structure at heterochromatic loci [8,9]. Biochemical studies have shown that the Fun30 from *S. cerevisiae* is a homodimer, possessing DNA-stimulated ATPase activity [10]. These studies have also shown that the purified protein can mediate nucleosome sliding as well as histone variant exchange; however, the efficiency in mediating histone variant exchange is greater than nucleosome sliding [10]. The protein has not yet been characterized in *C. albicans*.

In addition to ATP-dependent chromatin remodeling proteins, histone-modifying enzymes have also been shown to play a role in DNA repair. Rtt109 is a B-type histone acetyltransferase specific to fungi [11]. Studies have shown that the protein acetylates histone H3 at K56, K9, and K27 [12]. The H3K56ac modification is associated with the DNA damage response both in *S. cerevisiae* and in *C. albicans* [13,14]. This modification has been recently identified in mammalian cells too, where p300 has been shown to mediate the acetylation at the K56 position of histone H3 [15]. Studies have shown that Rtt109 and p300 do not share sequence homology but do share structural similarities [16]. Rtt109 in *C. albicans* has been linked to pathogenesis, as cells lacking Rtt109 were found to be sensitive to genotoxic agents such as camptothecin (CPT) and methyl methanesulfonate (MMS), indicating the role of the histone acetyltransferase in DNA damage response/repair [14]. Recently, we have shown that Rtt109 mediated acetylation of H3 regulates the expression of genes involved in GPI anchor biosynthesis and ergosterol biosynthesis [17].

*C. albicans* is an opportunistic pathogen causing candidiasis in immunocompromised patients. The organism is an intracellular pathogen, and the host cell combats the infection by generating ROS that causes DNA damage. Thus, *C. albicans* has evolved robust mechanisms to repair the damaged DNA thereby ensuring genomic stability [18]. As stated earlier, the role of Rtt109 but not of Fun30 has been delineated in *C. albicans* [14]. In addition, the crosstalk between Rtt109 and Fun30 has not yet been investigated.

In this paper, we have characterized the *C. albicans* Fun30 and also investigated the crosstalk between Fun30 and Rtt109. We show that *C. albicans* Fun30 is a functional homolog of both *S. cerevisiae* Fun30 as well as human SMARCAD1. The protein mediates both double-strand break end resection as well as regulating gene expression. In particular, *C. albicans* Fun30 regulates the expression of *RTT109* that catalyzes the acetylation of H3K56. Rtt109, in turn, regulates the expression of Fun30 via H3K56 acetylation (H3K56ac). Fun30 and H3K56ac together regulate the expression of *TEL1* and *MEC1* encoding for the sensor kinases Tel1, and Mec1, respectively. They also regulate the expression of *SNF2*, which encodes for another ATP-dependent chromatin remodeling protein. Finally, we show that Fun30 and Rtt109 regulate the expression of *RAD9*, encoding for a cell cycle checkpoint protein, as well as *RAD5*, encoding for an ATP-dependent chromatin remodeling protein in response to genotoxic stress, thus modulating the DNA damage response pathway on induction of DNA damage.

## 2. Materials and Methods

### 2.1. Chemicals

All chemicals used in this study were of analytical grade and were purchased either from Thermo Fisher Scientific (Waltham, MA, USA), Merck (Mumbai, Maharashtra, India), HiMedia (Mumbai, Maharashtra, India), SRL (Mumbai, Maharashtra, India), or Sigma-Aldrich (Burlington, MA, USA). The media for growing cells were purchased from HiMedia (Mumbai, Maharashtra, India). Restriction enzymes and T4 DNA ligase were purchased from Thermo Fisher Scientific (Waltham, MA, USA), Merck (Kenilworth, NJ, USA), and New England Biolabs (Ipswich, MA, USA). The TA cloning kit was purchased from MBI Fermentas (Waltham, MA, USA). The gel extraction kit was purchased from Qiagen (Hilden, Germany). SYBR Green was purchased from Kapa Biosystems (Basel, Switzerland). Bradford dye for protein estimation was purchased from Sigma-Aldrich (Burlington, MA, USA). All the primers were synthesized by GCC Biotech (Kolkata, West Bengal, India).

### 2.2. Plasmids and Strains

*E. coli* strain JM109 (Merck, Mumbai, Maharashtra, India) was used for cloning *RTT109*, *FUN30*, and *RAD5*. Two *C. albicans* strains, BWP17 and SN152, were used in this study, and all mutants were made in these backgrounds. The primers used for constructing the mutants are listed in Appendix A. The BWP17 was a kind gift from Prof. Aaron P. Mitchell, Department of Biological Sciences, Carnegie Mellon University, Pittsburgh, PA, USA. The SN152 strain was a kind gift from K. Natarajan, JNU, and the BWP17-*URA3* strain was a kind gift from S.S. Komath, JNU. The JKM179 strain was a kind gift from James Haber, University of Massachusetts. For complementation assays, *C. albicans FUN30* was cloned in a Yep*HIS* vector between BamHI and MluI restriction sites. The plasmids and strains used in this study are listed in Appendix A, respectively.

*C. albicans* strains were cultured in Yeast Extract-Peptone-Dextrose (YEPD) media or synthetic dextrose (SD) minimal media. Ura^−^ strains were cultured in SD media supplemented with 60 µg/mL uridine. His^−^ strains or Arg^−^ strains were cultured in SD medium supplemented with 85.6 μg/mL histidine or arginine. YEPD media were used to culture the yeast strain YPH500, while JKM179 was cultured in YEP-lactate and YEP-Galactose. Transformations were performed using the Lithium acetate method [19].

### 2.3. Antibodies

Antibodies of anti-H3 (Cat# 96C10) were purchased from Cell Signalling Technology (Danvers, MA, USA), anti-H3K56ac (Cat# 76307) was purchased from Abcam (Cambridge, UK), anti-c-Myc (Cat# ITG0001) was purchased from Immunotag (St. Louis, MO, USA), and anti-G6PDH (Cat# A9521) was purchased from Sigma-Aldrich (Burlington, MA, USA).

### 2.4. Generation of C. albicans and S. cerevisiae Mutant Strains

PCR-mediated gene disruption strategy was used for making all the mutants in *C. albicans* as well as in *S. cerevisiae.* Heterozygous mutants of *RTT109*, *FUN30*, and *VPS75* were made using the selection marker *HIS1*. The heterozygous mutant of *RAD9* was made using the selection marker *URA3*. For making the *RTT109* deletion mutant, the second copy of *RTT109* was replaced with an *ARG4* selection marker and transformed in *RTT109Hz* background. For *RTT109Hz/FUN30Hz* mutant, one copy of *RTT109* was replaced with *HIS1* followed by replacing one copy of *FUN30* with an *ARG4* selection marker. *RTT109Hz*/*FUN30*/*Hz*/*RAD9Hz-URA3* was created by replacing *RAD9* with a *URA3* selection marker in *RTT109Hz*/*FUN30Hz* background. The *HIS1*, *ARG4*, and *URA3* were amplified by PCR using primers that included flanking gene-specific sequences and transformed in the respective *C. albicans* strain. Selection of transformants was performed in SD media plate without Histidine or Arginine or Uridine and confirmed by PCR using gene-specific flanking primers.

The *FUN30* null mutant was made using a fusion PCR-based gene deletion strategy. The first round of reaction involved the amplification of flanking sequences of the target gene (with the genomic DNA as template and primer 1 and 3 or primer 4 and 6 in separate reactions) and the selectable marker (with pSN40 or pSN52 plasmid as template and primer 2 and 5) (primer sequences are provided in Appendix A). The 5’ overhangs of primer 2 and 4 are complementary to the 5’ overhangs of primer 3 and 5, respectively. In the second round, the “fusion PCR” reaction, all these three fragments were used as templates, and the fusion product was amplified using primers 1 and 6. A heterozygous mutant was first created by replacing one copy of *FUN30* with a *LEU2* marker. The null mutant was created in this background by replacing the remaining copy with the *HIS1* marker. Colonies were screened on Leu^−^His^−^minimal media after transformation using the electroporation technique.

The yeast strains YPH500 and JKM179 were used in this study. Sc*fun30∆* was generated by replacing *FUN30* with a *URA3* selection marker in the YPH500 strain. *fun30*∆ strain was similarly constructed in the JKM179 strain. *URA3* cassette was amplified using the pYES2 vector template. YPH500-*URA3* and JKM179-*URA3* strains were created by replacing a short intergenic region present downstream of the *FUN30* gene with a *URA3* selection marker. The Yep*HIS* vector was also transformed in the same strains. *Scfun30∆*-Yep*HIS and Scfun30∆-*Yep*HIS-CaFUN30* strains were generated by transforming the Yep*HIS* vector and Yep*HIS* containing the *CaFUN30* gene, respectively. YPH500-*URA3* and JKM179-*URA3* strains were selected on uridine minimal media, while *Scfun30∆*-Yep*HIS* and *Scfun30∆-*Yep*HIS-CaFUN30* were selected on Ura^−^Leu^−^ minimal media.

To understand the role of ATPase activity of Fun30 in transcriptional regulation of DNA damage responsive genes, *FUN30Hz*/*FUN30*OE^K593A^-*URA3* was generated by PCR-based site-directed mutagenesis in Motif Ⅰ (GLGK^593^T) of *FUN30* cloned in p*ACT1*-*GFP* vector. The vector was transformed in the *FUN30Hz* strain and selection of transformants was performed in SD media plate without Histidine and Uridine.

### 2.5. Generation of Overexpression Strains

One allele of both *RTT109* and *FUN30* were introduced into the *FUN30Hz* and *RTT109D* backgrounds, respectively. All the genes were PCR amplified using Phusion Plus DNA polymerase (Catalog # F630S; Themo Fisher Scientific, Waltham, MA, USA). One allele of *RAD5* was introduced into *RTT109Hz*/*FUN30Hz* as well as the BWP17 strain. For creating the overexpression strains, *RTT109*, *FUN30*, and *RAD5* were cloned under an *ACT1* promoter in the p*ACT1-GFP* vector. *RTT109* was cloned between the HindIII and NheI sites, while *FUN30* was cloned between the PstI and Nhe1 sites. *RAD5* was cloned into the p*ACT1-GFP* vector between the HindIII and NheI sites. The constructs generated were linearized using the StuI restriction enzyme and introduced into the respective backgrounds. The transformants were selected on a media plate without uridine. The successful transformants were screened by PCR, using a gene-specific forward primer (FP; Appendix A) and a locus-specific *RPS1* reverse primer (RP; Appendix A) to confirm the integration at the *RPS1* locus. *URA3* expressing control strains were made by digesting the p*ACT1-GFP* vector, which has a *URA3* promoter, with Stu1 and transforming it.

### 2.6. Generation of Revertant Strains

Revertant strains of *RTT109* and *FUN30* in *RTT109* deleted and *FUN30* heterozygous backgrounds, respectively, were performed similarly, as described above.

### 2.7. Epitope-Tagging of Endogenous FUN30

A pADH34 vector, a kind gift of Prof. Johnson, UCSF, was used to add a 13X-Myc tag to the C-terminus of endogenous *FUN30* of *C. albicans*. Briefly, the fusion PCR method was used to generate appropriate double-stranded DNA which was transformed into the SN152 strain. The DNA was integrated into the site by homologous recombination, and transformed colonies were selected using a nourseothricin selection media plate. This strain was termed *FUN30myc*, and all the mutants were made in this background. This strain was used as the control wild type for ChIP, qPCR, and plate assays.

### 2.8. Cloning of C. albicans FUN30 into a Mammalian Expression Vector

*C. albicans FUN30* was amplified using gene-specific primers and cloned between Xho1 and Xma1 restriction sites into the pEGFP-C2 mammalian expression vector. The cloned plasmid was named pEGFP-C2-*CaFun30*.

### 2.9. Functional Complementation Assays

To determine whether *C. albicans FUN30* can functionally complement *S. cerevisiae FUN30*, the YPH500 strain of *S. cerevisiae* was used. A spot assay was conducted for the YPH500-*URA3*-Yep*HIS*, *Scfun30∆*-Yep*HIS*, and *Scfun30∆*-Yep*HIS*-*CaFUN30* strains in the presence of 0.01%, 0.02%, and 0.03% MMS (Methyl methanesulfonate) and 1 µg/mL, 2 µg/mL, and 4 µg/mL CPT [20]. A total of 5 samples of five-fold serial dilution were prepared for all the strains and spotted on control plates (YEP-Gal) as well as DNA damaging agent-containing plates. The plates were incubated at 30 °C for 30 h. Images were taken for interpretation of the results.

To determine whether *C. albicans FUN30* can functionally complement human *SMARCAD1*, HeLa cells were grown to 60% confluency on coverslips in 6 well plates. After 18 h of seeding, they were transfected either with scrambled siRNA (Control) or with siRNA against *SMARCAD1* using a Lipofectamine 3000 (Thermo Fisher, Waltham, MA, USA) reagent as per the manufacturer’s protocol. After 24 h of siRNAs transfection, the cells were transfected either with a pEGFP-C2 vector or with pEGFP-C2-*CaFun30* plasmids. After 24 h of plasmids transfections, the cells were treated with 10 μM CPT for 5 h. After the desired treatment, cells were processed by washing 3–4 times with 1X PBS followed by fixing with a 1:1 mixture of methanol and acetone for 10 min and, subsequently, permeabilizing with 0.5% Triton X-100 in 1X PBS for 10 min. The cells were then washed with 1X PBS and blocked in 2% BSA at room temperature for 1 h. After washing with 1X PBS, the cells were incubated with the γ-H2AX primary antibody in 2% BSA at 4 °C for 16 h. The cells were then washed 3–4 times with 1X PBS and incubated with the TRITC-conjugated secondary antibody and Hoechst 33342 in 2% BSA at 37 °C for 30 min. The cells were then washed 3–4 times with 1X PBS, and the coverslips were mounted on slides and viewed using a confocal microscope (Nikon, Tokyo, Japan) under a 60X oil immersion objective.

### 2.10. DNA End Resection

Primary cultures for the JKM179-*URA3*-Yep*HIS*, *Scfun30∆*-Yep*HIS*, and *Scfun30∆*-Yep*HIS*-*CaFUN30* strains were grown in YEP-lactate at 30 °C. 2% of primary culture was inoculated in 2% of YEP-Gal media, for induction of HO-endonuclease [6]. Cells were incubated for 8 h at 30 °C. Subsequently, the cells were harvested by centrifugation at 5000 rpm for 5 min and the cell extract was prepared by vortexing the cells with glass beads and lysis buffer (100 mM Tris.Cl (pH 8.0), 50 mM EDTA, and 1% SDS). DNA was extracted by the phenol and chloroform method. The extracted DNA was diluted, and 5 ng DNA was used as the template for qPCR (primer list is provided in Appendix A). The transcript level for an independent locus from chromosome 2 (*PHO5*) was used for normalization.

### 2.11. Quantitative PCR (qPCR)

Cells were taken from the late log phase secondary culture to isolate the RNA. The cells were collected by centrifugation and then washed twice with DEPC-treated water. The cells were lysed by vortexing with glass beads. A TRIzol reagent (Qiagen, Hilden, Germany) was added to the cell pellet to extract the total RNA. The integrity and quantity of the isolated RNA was estimated using a NanoDrop Spectrophotometer (Thermo Fisher Scientific, Waltham, MA, USA). cDNA was prepared using 2 µg of total RNA using random hexamers and dNTPs (Thermo Fisher Scientific, Waltham, MA, USA). The reverse transcriptase and RNase inhibitor for cDNA synthesis were purchased from New England Biolabs, USA. The transcript levels of different genes were quantified by amplifying with qPCR primers (Appendix A) using the SYBR Green PCR master mix. *GAPDH* transcripts levels were used as an internal control for all the experiments in the case of *C. albicans*. In the case of yeast, *ALG9* was used as the internal control. The fold change was calculated using the delta-delta Ct method [21].

### 2.12. Chromatin Immunoprecipitation (ChIP) Assay

Briefly, cells from the late log phase secondary culture were harvested after fixing with formaldehyde (1% final concentration) for 20 min. Then, glycine (25 mM) was added to quench the formaldehyde, and the cells were incubated for 10 min. Following this step, the cells were treated with 5 U lyticase enzyme for 2 h. Subsequently, the cells were pelleted down, washed, and lysed in lysis buffer (50 mM HEPES, pH 7.5; 140 mM NaCl; 1 mM EDTA; 1% Triton X-100; 1 mM PMSF) with glass beads and vortexed. The supernatant was sonicated using a water bath sonicator (five cycles of 15 s ON and 15 s OFF each). After sonication, the chromatin of different mutants was incubated overnight with the respective antibodies. Then, protein G agarose beads were added for binding with the antibody. The beads were subsequently washed with lysis buffer, high-salt buffer (lysis buffer with 500 mM NaCl), wash buffer (10 mM Tris.Cl, pH 8.0; 250 mM LiCl; 1 mM EDTA; 0.5% NP-40), and Tris-EDTA buffer. Finally, elution buffer (1.0% SDS; 100 mM NaHCO_3_) was added to elute the protein-DNA complex, and the protein was digested using proteinase K. The DNA was extracted using phenol: chloroform mixture (1:1) and then precipitated with isopropanol. The ChIP samples were analyzed by qPCR using promoter region-specific primers (Appendix A).

### 2.13. Western Blots

Cells were taken from the 8 h secondary culture at 30 °C, and lysates were prepared in ice-cold lysis buffer (50 mM Tris.Cl, pH 7.5; 150 mM NaCl; 50 mM NaF; 1% Triton X-100; 0.1% SDS; 3 mM PMSF) using glass beads. The lysate was centrifuged at 16,000 RCF, and supernatant was taken to measure the protein concentration. The protein samples were separated by SDS-PAGE. The protein was then transferred from the gel to a PVDF membrane. Membrane blocking was performed with 5% skimmed milk in PBS for 1 h. The blot was then probed with the respective primary antibody at 4 °C overnight followed by washing three times with PBST (PBS + 0.05% Tween 20). The blots were incubated with appropriate secondary antibodies. Subsequently, the blots were washed thrice with PBST (10 min per wash) and developed using chemiluminescence.

### 2.14. Immunofluorescence

Cells from the secondary culture were fixed with 37% formaldehyde. After washing the cell pellet with 1X PBS, lyticase treatment was given to digest the cell wall. Triton X-100 was used to permeabilize the cells. The cells were resuspended in a blocking solution (1% BSA in 1X PBS). The primary c-Myc antibody was added in a dilution of 1: 100 to the cells and then incubated overnight at 4 °C. The cells were pelleted, washed with PBS, and resuspended in the appropriate secondary antibody solution. Hoechst dye was added to it, and incubation was continued for 30 min. The cells were pelleted, washed with PBS, and mounted on glass slides. The slides were analyzed under Nikon A1R confocal microscope at 100× magnification.

### 2.15. Growth Rate Analysis

A primary culture was grown in 10 mL YEPD media at 30 °C. 0.1 OD cells from the primary culture was added into the secondary culture media and incubated at 30 °C, 220 rpm. OD at A_600nm_ was monitored with the aliquots taken from secondary culture every 2 h until saturation was reached. The growth curve was obtained from the OD values, and the doubling time was calculated.

### 2.16. Plate Assays

Cells from the secondary culture of different mutants were measured at 0.1 OD at A_600nm_ and diluted accordingly with 0.9% saline. A total of five samples of five-fold serial dilution were prepared and spotted on control plates as well as different treated plates. The plates were incubated at 30 °C. Images were taken for the interpretation of results. All plate assays were performed three times, and the best representative image is shown.

### 2.17. Minimal Inhibitory Concentration 50 (MIC_50_)

The MIC assay was performed using the broth dilution method, as described in [22]. Briefly, the primary culture was grown at 30 °C overnight in YEPD media. The secondary culture was grown for 6 h at 30 °C, by inoculating the 2% of primary culture in YEPD media. In a 96-well microtiter plate, 100 µL of sterile YEPD media was added to 11 wells. A volume of 200 µL sterile media was added to the 12th well, which was taken as a negative control. In the first well, 200 µL of 2× of the starting concentration of the antifungal drug (MMS) was added, and then 2× serial dilution was performed by adding 100 µL of MMS containing medium from one well to the next up to the 10th well. The 11th well did not contain the drug and was considered as the positive control. An OD_600_ nm corresponding to 0.1 was prepared from the secondary culture in sterile YEPD media, and from this cell suspension, 50× final dilution was made in suitable media. A volume of 100 µL was added to all the wells except the negative control. The microtiter plate was incubated at 30 °C with mild shaking for 24 h, and then OD_600_ nm was measured using a multi-plate reader (Thermo Scientific, Waltham, MA, USA). The MIC_50_ was calculated at the concentration of the drug where 50% of inhibition of growth was seen.

### 2.18. Phylogenetic and Domain Analysis

Phylogenetic analysis of the uncharacterized Fun30 protein sequence from *C. albicans* was performed using www.phylogeny.fr/version2_cgi/ (accessed on 17 May 2018) using default settings. The protein sequences of different ATP-dependent chromatin remodeling proteins from various organisms belonging to different families were extracted from NCBI. The domain architecture of Fun30 from *C. albicans* was created using the https://prosite.expasy.org/mydomains/ (accessed on 16 December 2019) website in its default settings.

For phylogenetic analysis, orf19.6291 protein from *C. albicans*, Rad5 from *S. cerevisiae*, Rad16 from *S. cerevisiae*, Lodestar from *D. melanogaster*, Etl1 from *M. musculus* and SMARCAD1 from *H. sapiens*, Fun30 from *S. cerevisiae*, Fft1, Fft2, and Fft3 from *S. pombe*, Iswi from *D. melanogaster*, Snf2 from *S. cerevisiae*, BRG1, BRM, and CHD7 from *H. sapiens*, Mi-2 from *D. melanogaster*, CHD1 from *S. cerevisiae*, EP400 from *H. sapiens* and *M. musculus*, SWR1 from *S. cerevisiae* and *C. elegans*, Ino80 from *H. sapiens*, *S. cerevisiae* and *C. elegans*, ERCC6 from *H. sapiens*, ATRX from *H. sapiens*, Rad54 from *S. cerevisiae*, SMARCAL1 from *H. sapiens*, and Mot1 from *S. cerevisiae* were used.

### 2.19. Statistical Analysis

All qPCR and ChIP experiments are reported as average ± standard error of mean (SEM) of three independent (biological) experiments unless otherwise specified. Each independent experiment was performed as at least two technical replicates. The statistical significance (*p*-value) was calculated using a paired t-test available in Sigma Plot. The differences were considered significant at *p* < 0.05.

## 3. Results

### 3.1. The orf19.6291 Encodes Fun30, a Member of the ATP-Dependent Chromatin Remodeling Protein Family

*C. albicans* orf19.6291 is an uncharacterized protein annotated as Fun30 in the Candida Genome database [23]. Phylogenetic analysis showed that orf19.6291 is closely related to Fun30 from *S. cerevisiae* (Appendix A). Further, bootstrap values suggest that the orf19.6291 evolved along with Fft3, Fft2, SMARCAD1, and Etl1 (Appendix A).

Multiple sequence alignment showed 46.21% sequence identity between *S. cerevisiae* and *C. albicans* Fun30 protein, whereas the human SMARCAD1 showed 33.13% sequence identity with *C. albicans* Fun30. The RecA-like domain 1 of *S. cerevisiae* showed 70.24%, and human SMARCAD1 showed 54.76% sequence identity with *C. albicans* Fun30. On the other hand, the RecA-like domain 2 of *S. cerevisiae* showed 63.76%, and the human SMARCAD1 showed 50.68% sequence identity with *C. albicans* Fun30.

Sequence analysis showed that orf19.6291 contained the conserved helicase motifs as well as the CUE motif (Appendix A). The CUE motif that has been shown to recognize and bind to monoubiquitinated proteins [24] is present in the N-terminus of Fun30 in the case of *S. cerevisiae*. However, the CUE motif in orf19.6291, as in the case of Fft2, is present between the subdomains of the helicase motifs (Appendix A). The CUE motif of *Ca*Fun30, *Sc*Fun30, and SMARCAD1 is approximately 35 amino acid stretch and is characterized by conserved phenylalanine and proline residues separated by a defined number of amino acids from a conserved leucine residue. At a distance of 10 amino acids from the conserved phenylalanine-proline residues, *Ca*Fun30 and human SMARCAD1 possessed conserved leucine residue, whereas *Sc*Fun30 has a valine in this position. These conserved residues are known to bind ubiquitin. In *S. cerevisiae*, it is reported that the CUE motif assists in gene silencing of the mating-type locus (*HMR*), telomeres, and the rDNA repeats [8]. Based on these analyses, it was concluded that orf19.6291 encodes Fun30 in *C. albicans*.

### 3.2. Fun30 Localizes to the Nucleus

The ATP-dependent chromatin remodeling proteins localize in the nucleus to mediate their function. To understand the localization of Fun30 in *C. albicans*, one of the alleles of *FUN30* in the SN152 strain was myc-tagged at the C-terminus, as explained in the Methods section. This strain is hereafter referred to as *FUN30myc*. Using an anti-myc antibody, the localization of the protein was found to be in the nucleus (Appendix A).

### 3.3. C. albicans Fun30 Is a Functional Homolog of Both S. cerevisiae Fun30 and Human SMARCAD1

Next, we addressed whether Fun30 from C*. albicans* is a functional homolog of Fun30 from *S. cerevisiae*. For this experiment, *FUN30* of *S. cerevisiae* was deleted in yeast strain YPH500 using homologous recombination, creating a *Scfun30∆* strain. The *FUN30* gene from *C. albicans* was cloned (without codon optimization) into the Yep*HIS* vector, as explained in the Methods section, and transformed into a *Scfun30∆* strain, creating a *Scfun30∆* -Yep*HIS*-*CaFUN30* strain. The control strain YPH500 and the *Scfun30∆* strain were also transformed with an empty Yep*HIS* vector to eliminate any effect due to the vector. Plate assays showed that *Scfun30∆* was sensitive to MMS and CPT as compared to the wild type (Figure 1A). This sensitivity was ablated when *C. albicans FUN30* was overexpressed in the *Scfun30∆* strain (Figure 1A). This led us to conclude that the Fun30 protein from *C. albicans* can rescue the growth defect in the presence of genotoxic agents, and thus can functionally complement Fun30 of *S. cerevisiae*.

We also asked whether *C. albicans* Fun30 can functionally complement human SMARCAD1. To address this question, we used HeLa cells wherein *SMARCAD1* was transiently knocked down using siRNA (Appendix A). The percentage of cells with γH2AX foci increased when treated with 10 µM CPT for 5 h as compared to untreated cells, as well as cells transfected with scrambled siRNA (Figure 1B–D) (*p* = 0.03; comparison drawn between control (scrambled siRNA transfected cells) and si*SMARCAD1* cells). The increased percentage of cells with γH2AX foci indicated that the double-strand break repair pathway was impaired when SMARCAD1 expression was downregulated, as reported earlier [25]. The double-strand break repair pathway was restored when *C. albicans FUN30* was overexpressed (without codon optimization) in *SMARCAD1* downregulated cells, as indicated by the decreased percentage of cells with γH2AX foci (Figure 1B–D and Appendix A) (*p* = 0.01; comparison done between si*SMARCAD1* cells and si*SMARCAD1* + *CaFUN30* cells). This led us to conclude that *C. albicans* Fun30 can functionally complement the human SMARCAD1.

Thus, *C. albicans* Fun30 is a functional homolog of both *S. cerevisiae* Fun30 and human SMARCAD1.

In the case of plate assays, five-fold serial dilutions were prepared and spotted on respective plates. The plates were incubated at 30 °C and imaged after 30 h.

### 3.4. Fun30 Protein of C. albicans Mediates Double-Strand Break End Resection

Studies have shown that the Fun30 family of proteins mediate Double-Strand Break (DSB) end resection when double-strand breaks are introduced into DNA by genotoxic agents [6]. To determine whether the Fun30 protein of *C. albicans* can also mediate DSB end resection, we used the yeast strain JKM179, as double-strand breaks targeted to specific loci cannot be introduced in the genomic DNA of *C. albicans*.

The endogenous *FUN30* gene was deleted in JKM179, an *S. cerevisiae* strain using homologous recombination, creating *fun30∆*. Next, *C. albicans FUN30* was overexpressed in the strain *fun30∆*. HO endonuclease expression, and consequently double-strand breaks, were induced by growing the cells in the presence of galactose for 8 h. Cells were harvested and genomic DNA was isolated, as explained in the Methods. The DSB end resection occurring on chromosome 3 was monitored by qPCR using primers, as reported by Eapen et al. [6].

In the absence of galactose, HO endonuclease was not produced, and consequently DSBs were not formed (Figure 2A). In the presence of galactose, HO endonuclease was induced, leading to DSB. End resection of DSB was found to be reduced in *fun30∆* cells as compared to the wild-type JKM179 strain (Figure 2B). The end resection was restored when *C. albicans FUN30* was overexpressed in *fun30∆* cells, indicating that the Fun30 protein of *C. albicans* possesses the ability to mediate DSB end resection (Figure 2B).

The data is presented as an average ± s.e.m of three independent experiments. The transcript level for an independent locus from chromosome 2 (*PHO5*) was used for normalization.

### 3.5. FUN30 Mediates the Response to Genotoxic Stress

To understand whether *FUN30* is required for the viability of *C. albicans* we deleted both the copies of the gene. First, a heterozygote mutant was created where one copy of the gene was deleted in SN152 cells using *LEU2* as the selection marker. The heterozygous mutant was confirmed by PCR using *LEU2* cassette-specific primers as well as gene-specific primers. The second copy was deleted using *HIS1* as the selection marker. The homozygous null mutant was confirmed using both *LEU2* and *HIS1* cassette-specific primers as well as gene-specific primers (Appendix A). Downregulation of gene expression was confirmed by qPCR (Figure 3A).

Next, the response of *FUN30* null mutant (*∆FUN30*) to genotoxic stress agents was studied, and it was found that the null mutant was sensitive to MMS, CPT, and H_2_O_2_, indicating that the gene product is required for the cell to respond to genotoxic stress (Figure 3B). It needs to be noted that the heterozygous mutant did not show any phenotypic defect, indicating that the gene is haplosufficient.

### 3.6. Fun30 Regulates the Expression of RTT109, SNF2, TEL1, and MEC1 in C. albicans

Fun30, an ATP-dependent chromatin remodeling protein, mediates not only DNA end resection but also modulates transcriptional regulation [7,26,27,28]. Previously, we had shown that in mammalian cells the DNA damage response pathway is transcriptionally regulated by ATP-dependent chromatin remodeling proteins [29,30,31]. Therefore, we investigated whether Fun30 regulates genes involved in the DNA damage response pathway in *C. albicans*. Specifically, we sought to investigate the expression of *RTT109* that encodes for Rtt109, histone acetyltransferase that acetylates H3 at K56 and other positions, and *TEL1* and *MEC1* that encode for Tel1 and Mec1, respectively. These proteins are functional homologs of ATM and ATR, respectively. Additionally, we also analyzed the expression of *SNF2,* which is the functional homolog of *BRG1*. Rtt109 mediated acetylation of H3K56 plays a role in DNA repair, and thus in genome stability [13,14]. Tel1 and Mec1, like ATM and ATR, are kinases that are activated on DNA damage and transduce the signal to downstream effectors via phosphorylation [32]. In mammalian cells, BRG1 has been shown to not only modulate the expression of ATM and ATR but also to directly participate in DNA damage repair [29,33,34].

In the remaining sections, we explore whether the changes in the gene expression are relevant to the response that *C. albicans* mounts towards genotoxic stressors.

Quantitative PCR showed that the expression of *RTT109* (*p* = 0.00 for both mutants), *SNF2* (*p* = 0.00 for both mutants), *TEL1* (*p* = 0.00 for both mutants), and *MEC1* (*p* = 0.00 for both mutants) was downregulated in both the heterozygote (*FUN30Hz*) and null mutant (*∆FUN30*) strains as compared to the wild type strain (Figure 3A).

To show that Fun30 specifically regulates the expression of a subset of genes, we also analyzed the expression of three genes—*RDN18* (encodes for 18S ribosomal RNA), *EFB1* (encodes translation elongation factor-beta), and *UBC13* (encodes for the ubiquitin-conjugating enzyme)—and found their expression to be unchanged in *FUN30Hz* and *∆FUN30* as compared to the wild type strain (Figure 3C).

We also queried whether this regulation was organism-specific. Therefore, the expression of *RTT109*, *SNF2*, *TEL1*, and *MEC1* was analyzed in the yeast strains YPH500 and *Scfun30∆*. We found that, as compared to the wild type, the expression of *RTT109*, *SNF2*, *TEL1*, and *MEC1* was downregulated in *Scfun30∆* (Appendix A), suggesting that this regulation exists in *S. cerevisiae* also.

### 3.7. The ATPase Activity of Fun30 Is Needed for Transcriptional Regulation

To confirm that the gene expression was dependent on the Fun30 gene expression, we used an overexpressor strain created in the BWP17 background. In this background, one copy of *FUN30* was deleted, creating the *FUN30Hz* strain. An ectopic copy of *FUN30* was expressed in this background under the expression of *ACT1* promoter using *URA3* as the selection marker, creating a *FUN30Hz*/*FUN30OE-URA3*strain. The SN152 has a *URA* gene, and therefore could not be used for creating the overexpressor strain.

Gene expression analysis showed that *FUN30*, *RTT109*, *TEL1*, and *MEC1* were downregulated in the *FUN30Hz* strain as compared to the wild-type BWP17 strain (Figure 3D). Further, the expression was restored when *FUN30* was overexpressed (Figure 3D).

To understand whether the ATPase activity of Fun30 was essential for transcriptional regulation, the conserved lysine (593 aa) of the Walker motif, GKT, was mutated to alanine. This lysine is important for ATPase activity in Fun30 of *S. cerevisiae* [8]. The mutated protein (K593A) was overexpressed in the *FUN30Hz* strain, and the expression of genes was analyzed by qPCR. We found that the expression of *FUN30*, *RTT109*, *TEL1*, and *MEC1* remained downregulated when K593A was overexpressed in the *FUN30Hz* strain indicating ATPase activity is needed for transcriptional regulation (Figure 3D).

### 3.8. Fun30 Binds to the Promoter Regions of the DNA Damage Response Genes

To assess whether Fun30 directly regulates the expression of *RTT109*, *SNF2*, *TEL1*, and *MEC1* genes, ChIP assays were performed. As antibodies against Fun30 from *C. albicans* are not available, we tagged one copy of the *FUN30* gene with *myc*, creating the *FUN30myc* strain in the SN152 background. In this background, we deleted one copy of FUN30, creating *FUN30mycHz*.

First, we confirmed that *FUN30* (*p* = 3.17 × 10^−6^), *RTT109* (*p* = 7.55 × 10^−6^), *SNF2* (*p* = 1.25 × 10^−7^), *TEL1* (*p* = 4.17 × 10^−7^), and *MEC1* (*p* = 3.10 × 10^−9^) were downregulated and the expression of *RDN18*, *UBC13*, and *EFB1* was unaltered (Appendix A). Next, ChIP experiments were performed using anti-myc antibodies. Fun30myc occupancy was found to be reduced on the *RTT109* (*p* = 4.70 × 10^−3^), *SNF2* (*p* = 1.60 × 10^−9^), *TEL1* (*p* = 2.24 × 10^−6^), and *MEC1* (*p* = 1.03 × 10^−5^) promoters in the *FUN30mycHz* strain (Figure 3E). In addition, we also found Fun30myc occupancy on the *FUN30* promoter (Figure 3E). The occupancy of the Fun30myc was reduced in the *FUN30mycHz* strain, indicating that the protein was autoregulating its expression (Figure 3E). The reduced occupancy of Fun30myc on the *FUN30* promoter possibly accentuates the reduced expression of Fun30, because western blots showed that Fun30myc expression (probed by anti-myc antibody) was also downregulated (Appendix A).

We also probed the occupancy of H3K56ac on the promoter regions of *FUN30* (*p* = 7.98 × 10^−6^), *RTT109* (*p* = 6.02 × 10^−4^), *SNF2* (*p* = 1.00 × 10^−2^), *TEL1* (*p* = 3.34 × 10^−6^), and *MEC1* (*p* = 1.81 × 10^−4^), and found it to be reduced, even though the global levels of H3K56ac were not altered, indicating that the recruitment of this modified nucleosome to the promoter is affected in the *FUN30mycHz* mutant strain (Figure 3F; Appendix A).

To confirm that Fun30 and H3K56ac occupancy occur on specific gene promoters rather than the whole genome, we checked the occupancy of these two proteins in an intergenic region present in Chr5 as well as on the *GAPDH* promoter and found that the proteins were not present in these regions (Appendix A).

Therefore, we conclude that Fun30 specifically localizes to the promoter regions of the DNA damage response genes and regulates their expression.

### 3.9. H3K56ac Regulates the Expression of FUN30, SNF2, TEL1, and MEC1

The role of Rtt109 in DNA damage repair has been well-delineated in both *S. cerevisiae* and *C. albicans* [13,14]. As the above experiments showed that H3K56ac is present on the promoter of DNA damage response genes, and that overexpression of *RTT109* restored the expression of the genes including *FUN30*, we asked whether a feedback loop exists in *C. albicans* wherein H3K56ac catalyzed by Rtt109 regulates *FUN30* like the feedback loop between SMARCAL1 and BRG1 in mammalian cells [31]. The expression of *FUN30* along with *SNF2*, *TEL1*, and *MEC1* was analyzed in cells where both copies of *RTT109* were deleted to create a homozygous null mutant, *rtt109*/*rtt109* (hereafter termed as *RTT109D*), using PCR based strategies (Appendix A). A revertant strain, *rtt109*/*rtt109*/*pACT1-RTT109* (*RTT109D*/*RTT109*OE-*URA*3), was also created, wherein *RTT109* was cloned into a p*ACT1* vector and integrated into an RPS1 locus. As this strain in BWP17 was created using *URA3* as the marker, all comparisons were done using the BWP17-*URA3* strain created specifically for this purpose. In *FUN30myc*, the two copies of *RTT109* were deleted using *HIS1* and ARG4 markers (Appendix A).

Quantitative PCR showed that the expression of *FUN30*, *SNF2*, *TEL1*, and *MEC1* was downregulated in *RTT109D* as compared to the wild type, irrespective of the strain background (Figure 4A,B). This downregulation was specific, as *RDN18*, *UBC13*, and *EFB1* expression was unaltered (Appendix A). Overexpression of *RTT109* under the control of the *ACT1* promoter in the *RTT109D* background (*RTT109D*/*RTT109OE-URA3*) restored the expression level of these genes (Figure 4A). Western blots confirmed that H3K56ac levels were downregulated in the *RTT109D* mutant and restored in the *RTT109D*/*RTT109OE-URA3* strain (Figure 4C). Interestingly, Fun30 expression was also not observed in the *RTT109D* mutant (Figure 4D and Appendix A).

ChIP analysis showed that H3K56ac was present on the promoters of *FUN30*, *SNF2*, *TEL1*, and *MEC1* (Figure 4E). The occupancy of H3K56ac decreased in the *RTT109D* mutant but was restored in the *RTT109OE* strain (Figure 4E). Finally, occupancy of Fun30myc tagged protein was also found to be decreased on the promoters of the DNA damage response genes in the *RTT109D* mutant (Figure 4F).

Thus, from these experimental results, it was concluded that both H3K56ac, catalyzed by Rtt109, and Fun30 regulate the expression of each other as well as of *SNF2*, *TEL1*, and *MEC1*.

### 3.10. The Catalytic Activity of Rtt109 Is Essential for the Transcription Regulation

The catalytic activity of Rtt109 requires the help of two chaperones—Asf1 and Vps75 [35]. Therefore, we hypothesized that blocking the catalytic activity of Rtt109 by deleting *VPS75* would also result in the downregulation of *RTT109*, *FUN30*, *SNF2*, *TEL1*, and *MEC1* expression. To test the hypothesis, one copy of Vps75 was deleted (*VPS75*/*vps75* referred to hereinafter as *VPS75Hz*) using a PCR-based strategy in the BWP17 strain. Quantitative PCR confirmed that the expression of *RTT109*, *FUN30*, *SNF2*, *TEL1*, and *MEC1* was indeed downregulated (Figure 4G) in the mutant strain, confirming that the catalytic activity of Rtt109 was essential for the transcriptional regulation mediated by H3K56ac.

### 3.11. RTT109Hz/FUN30Hz Double Mutant Shows a Differential Response to Genotoxic Stress

The above results show that H3K56ac catalyzed by Rtt109 and Fun30 transcriptionally regulates the DNA damage response pathway. What is the significance of this gene regulation? Further, do H3K56ac and Fun30 function in the same pathway? To answer these questions, a double heterozygous mutant with one copy each of *RTT109* and *FUN30* was created both in a BWP17 and an SN152 background. The mutant created in BWP17 was termed as *RTT109Hz*/*FUN30Hz*. A similar mutant made in the *FUN30myc* strain (SN152 background) was termed as *RTT109Hz/FUN30mycHz*.

Quantitative PCR confirmed that the expression of *RTT109* (*p* = 1.97 × 10^−6^ for *RTT109Hz*/*FUN30mycHz*), *FUN30* (*p* = 1.27 × 10^−6^ for *RTT109Hz*/*FUN30mycHz*), *SNF2* (*p* = 2.84 × 10^−5^ for *RTT109Hz*/*FUN30mycHz*), *TEL1* (*p* = 6.59 × 10^−7^ for *RTT109Hz*/*FUN30mycHz)*, and *MEC1* (*p* = 1.00 × 10^−10^ for *RTT109Hz*/*FUN30mycHz)* was downregulated in both the *RTT109Hz*/*FUN30mycHz* and *RTT109Hz*/*FUN30Hz* strains (Figure 5A,B and Appendix A; qPCR data for *FUN30Hz* is shown in Figure 3 while the qPCR data for *RTT109Hz* is shown in Figure 4). In contrast, the expression of *RDN18*, *UBC13*, and *EFB1* was unaltered (Figure 5C). The ability of the *RTT109Hz*, *FUN30Hz*, *RTT109Hz*/*FUN30mycHz*, and *RTT109Hz*/*FUN30Hz* to grow in the presence of H_2_O_2_ (generates ROS and induces base modifications), CPT (results in double-strand breaks), and MMS (induces methylation of bases) was studied using plate (spot) assays. The heterozygous strains did not show a phenotypic difference in the presence of these genotoxic agents as compared to the wild-type strain, indicating haplosufficiency. However, as compared to the wild-type strain, both the *RTT109Hz*/*FUN30mycHz* and *RTT109Hz*/*FUN30Hz* mutants showed a differential response to genotoxic stress. The mutants showed no response to CPT (Figure 5D,E) but were sensitive to H_2_O_2_ (Figure 5D,E) and resistant to MMS (Figure 5D,E). This was in contrast to the sensitivity shown in the case of *∆FUN30* (See Figure 3B) as well as already reported for the *RTT109* null mutant [14].

To understand whether the resistance to MMS was statistically significant, MIC80 values were calculated. The MIC80 was 0.04% ± 0.008% (*p* = 0.004) for BWP17, 0.04% ± 0.005% (*p* = 0.004) for *RTT109Hz*, 0.04% ± 0.011% (*p* = 0.004) for *FUN30Hz*, and 0.41% ± 0.08% (*p* = 0.004) for *RTT109Hz*/*FUN30Hz*. Thus, the *RTT109Hz*/*FUN30Hz* showed 10-fold greater resistance to MMS as compared to the wild type strain.

Thus, the response of *RTT109Hz*/*FUN30mycHz* and the *RTT109Hz*/*FUN30Hz* strain appears to be dependent on the type of damage induced by the genotoxic agent. Further, the results indicate that the two genes might be working together in response to damage caused by genotoxic stressors.

### 3.12. Rtt109 and Fun30 in RTT109Hz/FUN30Hz Regulate the Transcription of DNA Damage Response Pathways Genes Differentially in Response to Different Types of Genotoxic Stressors

Hydrogen peroxide, CPT, and MMS activate the base excision repair, double-strand break repair, and mismatch repair pathways, respectively [36,37,38]. The Mec1 kinase is a central transducer in these pathways, wherein after activation by phosphorylation it phosphorylates Rad9 and Mrc1. Rad9 functions in the DNA damage checkpoint (DDC) while Mrc1 is important for the DNA replication checkpoint (DRC) [39]. Both proteins promote phosphorylation of Rad53, also known as effector kinase, by Mec1, thus activating the cell cycle checkpoint. In addition, Rad5, an ATP-dependent chromatin remodeling protein as well as an E3 ubiquitin ligase, is required for the bypass of replication forks through MMS-damaged DNA [40]. In *S. cerevisiae*, Fun30 and Rad5 cooperate to repair the stalled replication fork. In the absence of Rad5, Fun30 null mutants show resistance to MMS [20]. We found that the expression of *RAD9*, *MRC1*, and *RAD5* was downregulated in both the *RTT109Hz*/*FUN30mycHz* and *RTT109Hz*/*FUN30Hz* strains (Figure 5A,B).

Further, based on the plate assays, we postulated that in the presence of H_2_O_2_, *MEC1*, *RAD9*, and *MRC1* would be downregulated in the *RTT109Hz*/*FUN30Hz* strain as compared to the wild-type strain; in the presence of CPT, the expression of *MEC1*, *RAD9*, and *MRC1* would be unchanged between the wild type and the mutant strain; and in the presence of MMS, the expression of *RAD5* would be downregulated leading to resistant phenotype.

Therefore, the expression of *MEC1*, *RAD9*, *MRC1*, and *RAD5*, along with *FUN30*, *RTT109*, *SNF2*, and *TEL1*, was evaluated in the mutant strain (*RTT109Hz*/*FUN30mycHz* as well as *RTT109Hz*/*FUN30Hz*) and compared with the wild-type strain (*FUN30myc* and BWP17 respectively) in the presence of these genotoxic stressors.

Quantitative PCR showed that in the presence of H_2_O_2_, the expression of *FUN30*, *RTT109*, *SNF2*, *TEL1*, *MEC1*, *RAD9*, *MRC1*, and *RAD5* was downregulated in the mutant strain as compared to the wild-type strain, thus resulting in a sensitive phenotype (Figure 6; Appendix A).

In contrast, in the presence of CPT, *MEC1* and *RAD9* expression was unchanged in the mutant strain as compared to the wild-type strain (Figure 6; Appendix A). Interestingly, some strain-specific differences were observed. The expression of *FUN30*, *RTT109*, *SNF2*, *TEL1*, *RAD5*, and *MRC1* was unchanged in *RTT109Hz*/*FUN30mycHz* (Figure 6). However, the expression of *FUN30*, *RTT109*, and *SNF2* was downregulated, while *TEL1* was upregulated in *RTT109Hz*/*FUN30Hz* (Appendix A). As the *MEC1* expression was unchanged in the mutant strains, we postulate that the DNA damage response is operational in the mutant strain in the presence of CPT, and therefore no phenotypic response was observed to this stress agent.

In the presence of MMS, expression of *MEC1* and *RAD9* was upregulated, but the expression of *MRC1* and *RAD5* was downregulated in the mutant strain as compared to the wild-type strain (Figure 6; Appendix A). Once again, strain-specific differences were observed as *SNF2* and *TEL1* were upregulated in *RTT109Hz*/*FUN30mycHz* but were downregulated in *RTT109Hz*/*FUN30Hz*.

We also evaluated gene expression in the *∆FUN30* mutant and found that the gene expression of *FUN30*, *RTT109*, *SNF2*, *TEL1*, *MEC1*, *RAD9*, *MRC1*, and *RAD5* remained downregulated in the presence of genotoxic stressors (Appendix A). Thus, the differential regulation is only observed when limited amounts of Rtt109 and Fun30 proteins are expressed.

In conclusion, our data show that Rtt109 and Fun30 in *RTT109Hz*/*FUN30Hz* epigenetically regulate the DNA damage response pathway depending on the type of DNA damage induced by the genotoxic stress. Further, a cell lacking the *FUN30* gene but possessing both copies of *RTT109* shows sensitivity to genotoxic stressors because the DNA damage response genes remain downregulated.

### 3.13. The Response of RTT109Hz/FUN30Hz to DNA Damage Is Regulated by the Occupancy of Fun30 and H3K56ac on RAD9, MRC1, and RAD5 Promoter

To understand how Rtt109 and Fun30 epigenetically regulate the DNA damage response pathway, ChIP experiments were performed to study the occupancy of H3K56ac and Fun30 on the promoters of *FUN30*, *RTT109*, *SNF2*, *TEL1*, *MEC1*, *RAD9*, *MRC1*, and *RAD5* genes in *FUN30myc*, and *RTT109Hz*/*FUN30mycHz* mutant strain. In untreated cells, the occupancy of both Fun30myc and H3K56ac decreased on these gene promoters in the *RTT109Hz*/*FUN3mycHz* strain as compared to the *FUN30myc* wild type strain (Figure 7A,B and Appendix A). The decreased occupancy correlates with the decrease in expression of these genes in the mutant strain as compared to the wild-type strain.

When cells were treated with H_2_O_2_, the occupancy of Fun30 decreased on *FUN30*, *RTT109*, *MEC1*, *MRC1*, and *RAD5* promoters in the mutant strain as compared to the wild-type strain (Figure 7C and Appendix A). The H3K56ac occupancy did not alter between the mutant and wild-type strains on these promoters (Figure 7D and Appendix A), indicating that Fun30 is the primary driver for the alteration of gene expression observed during oxidative stress. Thus, *FUN30* but not *RTT109* plays an important role in the response the cell mounts to oxidative stress.

On treatment with CPT, as expected, the occupancy of Fun30 and H3K56ac was similar in the wild type *FUN30myc* and mutant strain on all promoters examined, leading to no change in the gene expression between the two strains in the presence of this genotoxic agent (Figure 7E,F and Appendix A).

Finally, when cells were treated with MMS, the occupancy of Fun30 and H3K56ac increased on the *RAD9* promoter in the mutant strain as compared to the wild-type strain (Figure 7G,H and Appendix A). Concomitantly, the occupancy of both Fun30 and H3K56ac decreased on *MRC1* and *RAD5* promoters in the mutant strain as compared to the wild-type strain (Figure 7G,H and Appendix A) leading to increased expression of *RAD9* and decreased expression of *MRC1* and *RAD5*. Thus, on treatment with an alkylating agent, both Fun30 and H3K56ac epigenetically modulate the expression of the DNA damage response genes.

### 3.14. Single Copy Deletion of RAD9 in RTT109Hz/FUN30Hz Rescues the Resistance to MMS

In presence of MMS, *RAD9* is upregulated while *RAD5* and *MRC1* are downregulated in *RTT109Hz*/*FUN30Hz* as well as *RTT109Hz*/*FUN30mycHz* as compared to the wild type strains.

In *S. cerevisiae*, deletion of both Rad5 and Fun30 results in resistance to MMS that can be rescued by overexpression of *RAD5* [20]. To understand whether this pathway is also operational in *C. albicans*, *RAD5* was ectopically overexpressed in the *RTT109Hz*/*FUN30Hz* mutant to generate the *RTT109Hz*/*FUN30Hz*/*RAD5OE*-*URA3* strain. The *URA3* gene was incorporated in both the BWP17 and *RTT109Hz*/*FUN30Hz* strains, creating the BWP17-*URA3* and *RTT109Hz*/*FUN30Hz*-URA3 strains for comparison purposes. In addition, *RAD5* was also overexpressed in BWP17 (*RAD5*OE-*URA3*) to determine the effect of overexpression of this gene on growth and drug sensitivity in normal cells.

Quantitative PCR indicated that the *RAD5* transcript was overexpressed in the *RTT109Hz*/*FUN30Hz*/*RAD5OE-URA3* strain as well as in *RAD5OE-URA3* (Figure 8A,B). Plate assays showed that the *RTT109Hz*/*FUN30Hz*/*RAD5OE*-*URA3* mutant was slow-growing as compared to the BWP17-*URA3* and *RTT109Hz*/*FUN30Hz-URA3* strains in YEPD plate in the absence of MMS (Figure 8C). However, *RAD5OE*-*URA3* did not show any growth difference as compared to BWP17-*URA3* (Figure 8D). In the presence of MMS, the growth of the *RTT109Hz*/*FUN30Hz*/*RAD5OE-URA3* was slow as compared to the *RTT109Hz*/*FUN30Hz-URA3* mutant, but it was impossible to determine whether this slow growth was because of the rescue of the phenotype or because of the inherent slow growth of the mutant strain (Figure 8E). It needs to be noted that *RAD5OE*-*URA3* showed greater sensitivity to MMS as compared to BWP17-*URA3* (Figure 8F). Therefore, duplication time was calculated for the wild type and the mutant strain in the absence and presence of MMS. In the absence of MMS, *RTT109Hz*/*FUN30Hz*/*RAD5OE-URA3* showed 4-fold slower growth as compared to the wild-type strain (Figure 8G and Appendix A). However, in the presence of MMS, the growth of *RTT109Hz*/*FUN30Hz*/*RAD5OE-URA3* was slower as compared to the *RTT109Hz*/*FUN30Hz-URA3* mutant but faster than BWP17-*URA3*, suggesting that *RAD5* overexpression can indeed partially rescue the resistant phenotype of *RTT109Hz*/*FUN30Hz* to MMS (Figure 8H and Appendix A).

Next, the role of *RAD9* was investigated. One copy of *RAD9* was deleted, creating *RTT109Hz*/*FUN30Hz*/*RAD9Hz-URA3*. A control strain *RAD9Hz-URA3* was also created in the BWP17 background by deleting one copy of the gene. Quantitative PCR confirmed that the expression of the gene was downregulated in both the *RTT109Hz*/*FUN30Hz*/*RAD9Hz-URA3* and *RAD9Hz-URA3* strains (Figure 8I,J). Plate assays showed that *RTT109Hz*/*FUN30Hz*/*RAD9Hz-URA3* and *RAD9Hz-URA3* showed growth similar to the BWP17-*URA3* strain on the YEPD plate (Figure 8C,D). Finally, in the presence of MMS, *RTT109Hz*/*FUN30Hz*/*RAD9Hz-URA3* showed sensitivity like the BWP17-*URA3* strain (Figure 8E). Growth analysis confirmed that the duplication time, as well as the growth curve of *RTT109Hz*/*FUN30Hz*/*RAD9Hz-URA3* and BWP17-*URA3*, are indeed similar, indicating that *RAD9* overexpression is the main reason for the resistance of *RTT109Hz*/*FUN30Hz* in the presence of MMS (Figure 8G,H; Appendix A).

## 4. Discussion

DNA damage induced by genotoxic stress activates the DNA damage response pathway. The pathway senses the DNA damage and transduces the message to the DNA damage repair proteins as well as cell checkpoint proteins to initiate DNA damage repair and cell cycle arrest. The Tel1 and Mec1 kinases belonging to the PIKK family are the central transducers of DNA damage [32]. These kinases are activated by autophosphorylation and in turn phosphorylate the effector proteins. Amongst these are the mediator proteins Rad9 and Mrc1, that on activation by phosphorylation transduce the signal to Rad53 [39,41,42]. Rad53, in turn, induces cell cycle arrest [43,44]. Studies have shown that Rad9 activates cell cycle arrest in G1, S, and G2/M phases, while Mrc1 predominantly functions during the S phase [41,45]. Further, Rad9 is primarily involved in the double-strand break pathway, while Mrc1 is involved when replicating DNA is damaged. In addition to Rad9 and Mrc1, Rad5, an ATP-dependent chromatin remodeling possessing E3 ubiquitination activity is involved in the replication bypass mechanism when the bases are modified by alkylating agents [40]. Studies have also shown that in addition to its role in the activation of cell cycle checkpoint activation, Rad9 is also required for transcription regulation of DNA damage response proteins [46].

Although extensive studies have shown that the DNA damage response and the DNA damage repair pathways are regulated by post-transcriptional modification, much less information is available about the transcriptional regulation of these pathways.

In this paper, we show that Fun30, an ATP-dependent chromatin remodeling protein, and histone acetylation catalyzed by Rtt109, a B-type histone acetyltransferase, transcriptionally regulate the response *C. albicans* mounts in the presence of genotoxic stress.

Phylogenetic analysis showed that *C. albicans* Fun30 is closely related to *S. cerevisiae* Fun30, with the two proteins showing 46.21%. One of the interesting deviations was in the placement of the CUE motif. In *S. cerevisiae*, this motif is present in the N-terminus, whereas in the *C. albicans* it is present between RecA- like domains 1 and 2. The CUE motif recognizes monoubiquitination and has been shown to assist in Fun30 activity in *S. cerevisiae* [8,24]. In the case of human SMARCAD1, the CUE motif has been shown to mediate interaction with the KRAB-associated protein KAP1 [47]. The role of the CUE motif in *C. albicans* Fun30 still needs to be delineated.

Fun30 of *C. albicans*, like its homologs in other organisms, possesses the ability to mediate DSB end resection, indicating functional conservation. Further, as in *S. cerevisiae* [8,48], *FUN30* is not essential for viability in *C. albicans*. The gene is required for mounting a response to genotoxic stressors.

Quantitative PCR and ChIP studies showed that the Fun30 protein in *C. albicans* regulates the expression of *RTT109*, *TEL1*, *MEC1*, *SNF2*, *RAD9*, *MRC1*, and *RAD5*. It is also present on its promoter, possibly regulating itself.

*RTT109* is non-essential, and the protein encoded by this gene catalyzes the formation of acetylated histones [13,14]. ChIP studies showed that H3K56ac is present on the promoters of *FUN30*, *RTT109*, *TEL1*, *MEC1*, *SNF2*, *RAD9*, *MRC1*, and *RAD5*; the occupancy of H3K56ac decreases on these promoters when both copies of *RTT109* are deleted correlating with decreased expression.

Thus, Fun30 and Rtt109 both regulate the expression of each other as well as of other DNA damage response genes.

The double heterozygous mutant *RTT109Hz*/*FUN30Hz* helped to elucidate how these two genes regulate the expression of the DNA damage response genes in the presence of different genotoxic agents. *RTT109Hz*/*FUN30Hz* shows sensitivity to H_2_O_2_, no response to CPT, and resistance to MMS as compared to the wild-type cells. qPCR data showed that in the presence of CPT, which induces double-strand break, the expression of the DNA damage response genes is upregulated in *RTT109Hz/FUN30Hz*, indicating that the remaining one copy of the two genes is either sufficient for the process or another set of proteins is activated to ensure that the cell repairs the double-strand breaks.

The *RTT109Hz*/*FUN30Hz* mutant is unable to cope with the induction of oxidative stress as the expression of *MEC1*, *RAD9*, *MRC1*, and *RAD5* is downregulated. This downregulation is primarily mediated by the absence of Fun30, as the occupancy of H3K56ac in the mutant is like the wild type. However, it needs to be remembered that Rtt109 null mutants are sensitive to oxidative stress [14]. Therefore, genome-wide occupancy studies would need to be conducted to understand the role of H3K56ac in driving gene expression during oxidative stress.

ChIP analysis shows that both Fun30 and H3K56ac regulate the expression of *RAD5* as well as *RAD9* in the double heterozygous mutants. In the mutant cells, on treatment with MMS, the occupancy of Fun30 and H3K56ac on the promoter of *RAD5* decreases, leading to reduced expression of the Rad5 protein. On the other hand, the occupancy of Fun30 and H3K56ac increased on the *RAD9* promoter correlating with increased expression of this gene.

When cells are treated with MMS, the DNA bases (G and A, primarily) are methylated. Methylation of bases results in the stalling of the replication fork, and Rad5 plays an important role in the bypassing of the stalled replication fork, either by recruiting DNA pol η (TLS pathway) or by polyubiquitinating PCNA and bypassing the replication fork [49]. By a mechanism possibly involving Rad51 and HR mediated repair, deletion of both Fun30 and Rad5 in *S. cerevisiae* has been shown to result in resistance to MMS [20]. In *RTT109Hz*/*FUN30Hz* mutant cells, the expression of *RAD5* and *MRC1* is downregulated but not completely abolished. It is possible that reduced expression of *RAD5* in the double heterozygous results in the bypass TLS pathway leading to faster DNA replication and thus more growth. However, FACS studies will need to be conducted to confirm this hypothesis. It also needs to be noted that overexpression of *RAD5* only partially restores sensitivity to MMS, indicating that the reduced expression of this gene might not be the primary driver for resistance to MMS in the *RTT109Hz*/*FUN30Hz* strain.

Concomitantly, the expression of *RAD9* is upregulated in *RTT109Hz*/*FUN30Hz* cells on treatment with MMS. Deletion of one copy of *RAD9* in *RTT109Hz*/*FUN30Hz* led to the restoration of a sensitive phenotype in the presence of MMS. Rad9 is known to inhibit DNA end resection, and studies in *S. cerevisiae* have shown that Fun30 inhibits Rad9 in promoting DNA end resection during DNA double-strand break repair [50]. Both Fun30 and Rad9 are known to play a role in resistance to MMS; however, the crosstalk between *RTT1109*, *FUN30*, and *RAD9* in *C. albicans* in determining the response to MMS needs to be further studied.

Epigenetic regulation of the genes involved in the DNA damage response/repair pathway has been previously reported. SMARCAL1 and BRG1, for example, have been shown to regulate the expression of *ATM*, *ATR*, *DROSHA*, *DICER*, and *DGCR8* in HeLa cells treated with doxorubicin. This regulation is important for the synthesis of non-coding RNA, and therefore for the recruitment of 53BP1 to the site of DNA damage [29,30,31]. DNA methylation has been shown to regulate the expression of mismatch repair genes [51].

The major question that the study reported in this paper raises is as follows: how do Fun30 and Rtt109 regulate the same set of genes differentially in the presence of different genotoxic stress? One possibility is the formation of specific protein complexes. The epigenetic regulators form many different protein complexes and are known to be recruited to the same set of genes under different conditions. For example, distinct SWI/SNF complexes have been shown to mediate transcriptional regulation of genes in HepG2 cells, with the expression of a given gene being a product of interaction between the complexes and other proteins [52]. Thus, we hypothesize that distinct protein complexes must be formed in the presence of different genotoxic stresses, leading to differential expression of genes involved in the DNA damage response pathway.

Thus, our studies show that the DNA damage response, as well as the repair pathway, are epigenetically modulated in *C. albicans*. The activation of the DNA damage response pathway needs to be regulated such that when the repair is completed the pathway is shut off. In mammalian cells, this feedback regulation is mediated by phosphorylated ATM such that the transcriptional activation occurs until phosphoATM is present in the cell [29]. On dephosphorylation, the transcription is repressed, and the DNA damage response pathway is switched off. It still needs to be investigated whether such feedback regulation exists in *C. albicans*.

## Figures and Tables

**Figure 1 jof-08-00559-f001:**
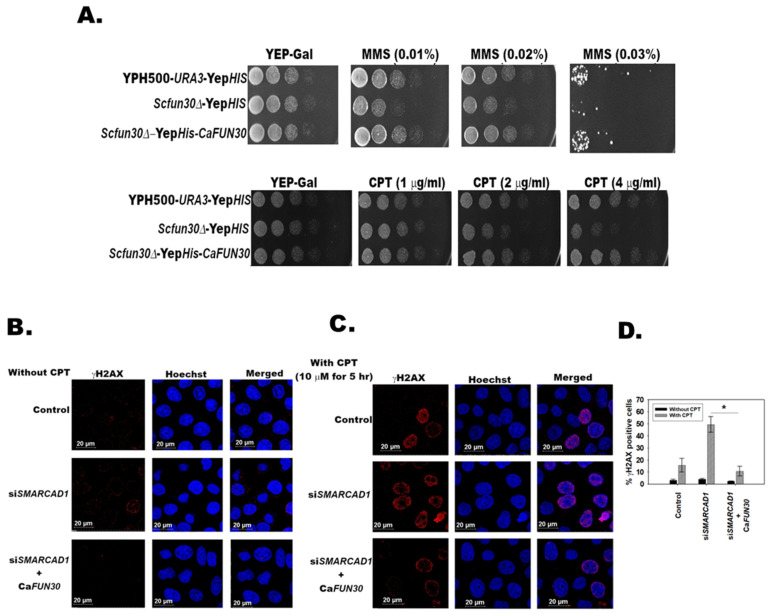
*C. albicans* Fun30 is a functional homolog of both *S. cerevisiae* Fun30 and human SMARCAD1. (**A**). Sensitivity to MMS and CPT of wild type YPH500 *URA3* containing Yep *HIS* (YPH500-*URA3*-Yep *HIS*), *Scfun30∆-*Yep*HIS*, and *Scfun30∆* overexpressing *C. albicans FUN30* (*Scfun30∆-*Yep*HIS-CaFUN30*) was studied using plate (spot) assays. (**B**). γH2AX foci was monitored using confocal microscopy in HeLa cells without CPT treatment. (**C**). γH2AX foci was monitored using confocal microscopy in HeLa cells after treatment with 10 µM CPT for 5 h. (**D**). Quantitation of γH2AX positive cells. Control indicates HeLa cells transfected with control scrambled siRNA. The data is presented as an average ± s.d. of two independent experiments where at least 100 cells were counted in each experiment. * *p* < 0.05. In panels B and C, Red is γH2AX and Blue is DNA stained with Hoechst.

**Figure 2 jof-08-00559-f002:**
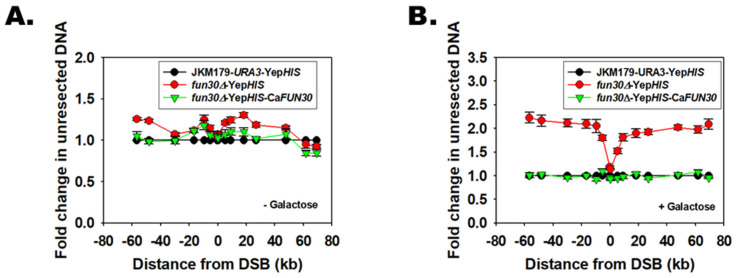
Fun30 protein of *C. albicans* mediates double-strand break end resection. qPCR was performed on genomic DNA isolated from wild type (JKM179-*URA3*-Yep*HIS*), *fun30∆* Yep*HIS*, and *fun30∆* overexpressing Ca*FUN30* (*fun30∆*-Yep*HIS*-Ca*FUN30*) cells at the indicated distances from the DSB in the (**A**). absence of galactose, and (**B**). presence of galactose.

**Figure 3 jof-08-00559-f003:**
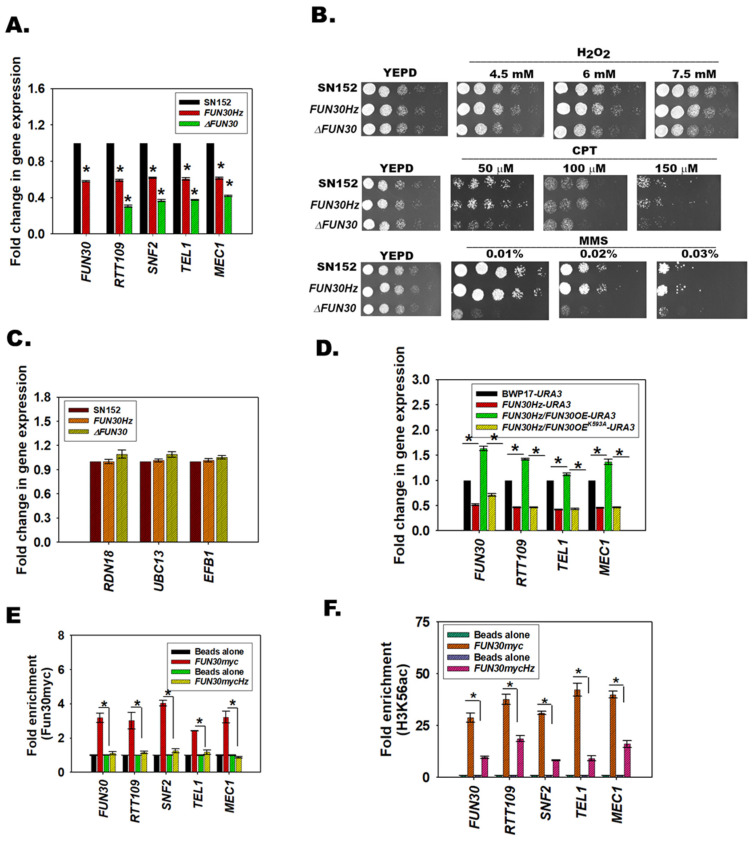
Fun30 regulates the expression of *RTT109*, *SNF2*, *TEL1*, and *MEC1* in *C. albicans*. (**A**). Expression of *FUN30*, *RTT109*, *SNF2*, *TEL1*, and *MEC1* in *FUN30Hz* and *∆FUN30* was compared to the expression in the wild-type strain using qPCR. (**B**). Plate assays were performed in response to genotoxic agents. (**C**). Expression of *RDN18*, *UBC13,* and *EFB1* in *FUN30Hz* and *∆FUN30* was compared to the expression in the wild-type strain using qPCR. (**D**). Expression of *FUN30*, *RTT109*, *TEL1*, and *MEC1* in wild type BWP17-*URA3, FUN30Hz*-*URA3*, *FUN30Hz/FUN30OE-URA3*, and *FUN30Hz/FUN30OE^K593A^-URA3* strains using qPCR. The *FUN30OE^K593A^* construct overexpresses an ATPase-dead mutant of Fun30 protein. (**E**). The occupancy of Fun30myc on *FUN30, RTT109, SNF2, TEL1,* and *MEC1* promoters was compared in *FUN30myc* and *FUN30mycHz* strains by ChIP. (**F**). The occupancy of H3K56ac on *FUN30, RTT109, SNF2, TEL1,* and *MEC1* promoters was compared in *FUN30myc* and *FUN30mycHz* strains by ChIP. Unless stated otherwise, the qPCR and ChIP data are presented as average ± s.e.m. of three biological replicates. A star indicates *p* < 0.05.

**Figure 4 jof-08-00559-f004:**
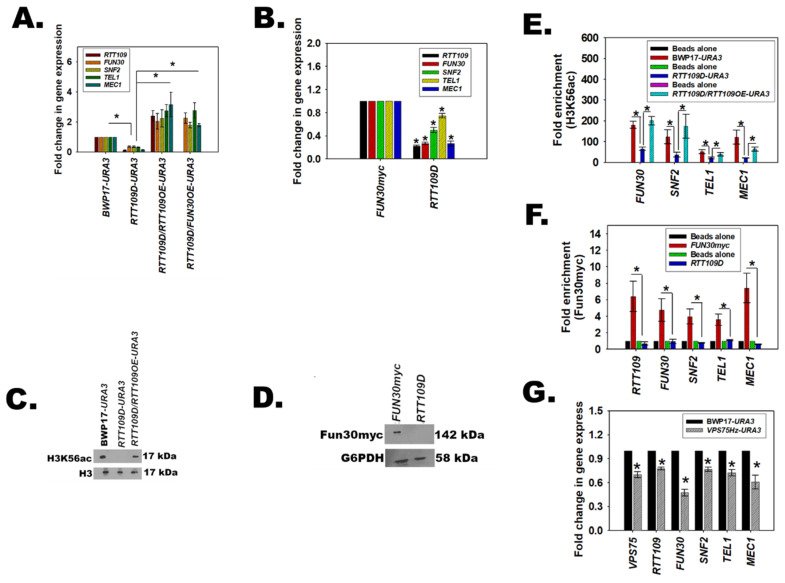
H3K56ac regulates the expression of *FUN30*, *SNF2*, *TEL1*, and *MEC1*. (**A**). Expression of *RTT109*, *FUN30*, *SNF2*, *TEL1*, and *MEC1* was analyzed in BWP17-*URA3*, *RTT109D*-*URA3*, *RTT109D*/*RTT109OE-URA3*, and *RTT109D*/*FUN30OE-URA3* using qPCR. (**B**). Expression of *RTT109*, *FUN30*, *SNF2*, *TEL1*, and *MEC1* was analyzed in *FUN30myc* and *RTT109D* strains using qPCR. This data is presented as average ± s.e.m. of three technical replicates. (**C**). Expression of H3K56ac was analyzed by western blot in BWP17*-URA3*, *RTT109D*-*URA3*, *RTT109D*/*RTT109OE-URA3* strains. H3 was used as the internal control. (**D**). Expression of Fun30myc was analyzed by western blot *FUN30myc* and *RTT109D* strains. G6PDH was used as the internal control. (**E**). Occupancy of H3K56ac on *FUN30*, *SNF2*, *TEL1*, and *MEC1* promoters was analyzed using ChIP in BWP17-*URA*3, *RTT109D*-*URA3*, *RTT109D/RTT109OE-URA3* strains. (**F**). Occupancy of Fun30myc on *RTT109*, *FUN30*, *SNF2*, *TEL1*, and *MEC1* promoters was analyzed using ChIP in *FUN30myc* and *RTT109D* strains. (**G**). Expression of *VPS75*, *RTT109*, *FUN30*, *SNF2*, *TEL1* and *MEC1* was analyzed in BWP17-*URA3* and *VPS75Hz-URA3* strains by qPCR. Unless stated otherwise, the qPCR and ChIP data are presented as average ± s.e.m. of three biological replicates. * *p* < 0.05. The western blots were repeated for three biological replicates and one representative image is shown.

**Figure 5 jof-08-00559-f005:**
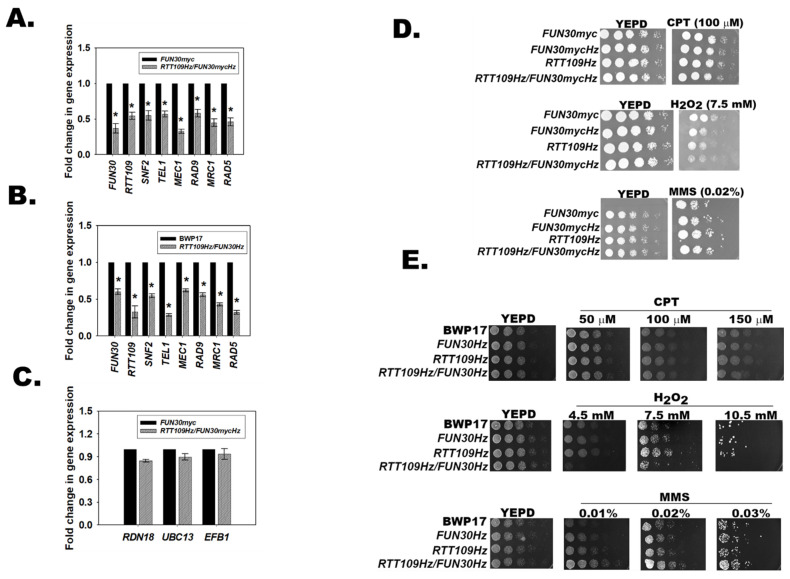
*RTT109Hz*/*FUN30Hz* double mutant shows a differential response to genotoxic stress. (**A**). Expression of *FUN30*, *RTT109*, *SNF2*, *TEL1*, *MEC1*, *RAD9*, *MRC1*, and *RAD5* was evaluated in *FUN30myc* and *RTT109Hz*/*FUN30mycHz* strains using qPCR. (**B**). Expression of *FUN30*, *RTT109*, *SNF2*, *TEL1*, *MEC1*, *RAD9*, *MRC1*, and *RAD5* was evaluated in BWP17 and *RTT109Hz/FUN30Hz* strains using qPCR. (**C**). Expression of *RDN18*, *UBC13*, and *EFB1* was analyzed by qPCR in *RTT109Hz*/*FUN30mycHz* strain. (**D**). Sensitivity of *FUN30myc*, *RTT109Hz*, *FUN30Hz*, and *RTT109Hz/FUN30mycHz* strains was studied in the presence of CPT (100 µM), H_2_O_2_ (7.5 mM), and MMS (0.02%). (**E**). Sensitivity of BWP17, *RTT109Hz*, *FUN30Hz*, and *RTT109Hz/FUN30Hz* strains was studied in the presence of CPT, H_2_O_2_, and MMS. The qPCR data are presented as average ± s.e.m. of three biological replicates. * *p* < 0.05. In the case of plate assays, five-fold serial dilutions were prepared and spotted on respective plates. The plates were incubated at 30 °C and imaged after 24 h.

**Figure 6 jof-08-00559-f006:**
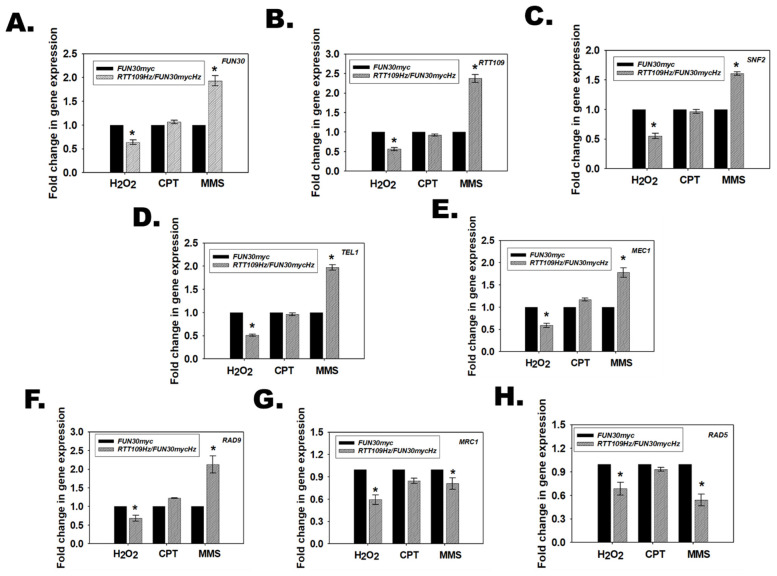
Rtt109 and Fun30 in *RTT109Hz*/*FUN30Hz* regulate the transcription of DNA damage response pathways genes differentially in response to different types of genotoxic stressors. Expression of (**A**). *FUN30*; (**B**). *RTT109*; (**C**). *SNF2*; (**D**). *TEL1*; (**E**). *MEC1*; (**F**). *RAD9*; (**G**). *MRC1*; (**H**). *RAD5* was analyzed in *FUN30myc* and *RTT109Hz*/*FUN30mycHz* strains in the presence of H_2_O_2_ (7.5 mM), CPT (100 µM), and MMS (0.02%). The qPCR data are presented as average ± s.e.m. of three biological replicates. * *p* < 0.05.

**Figure 7 jof-08-00559-f007:**
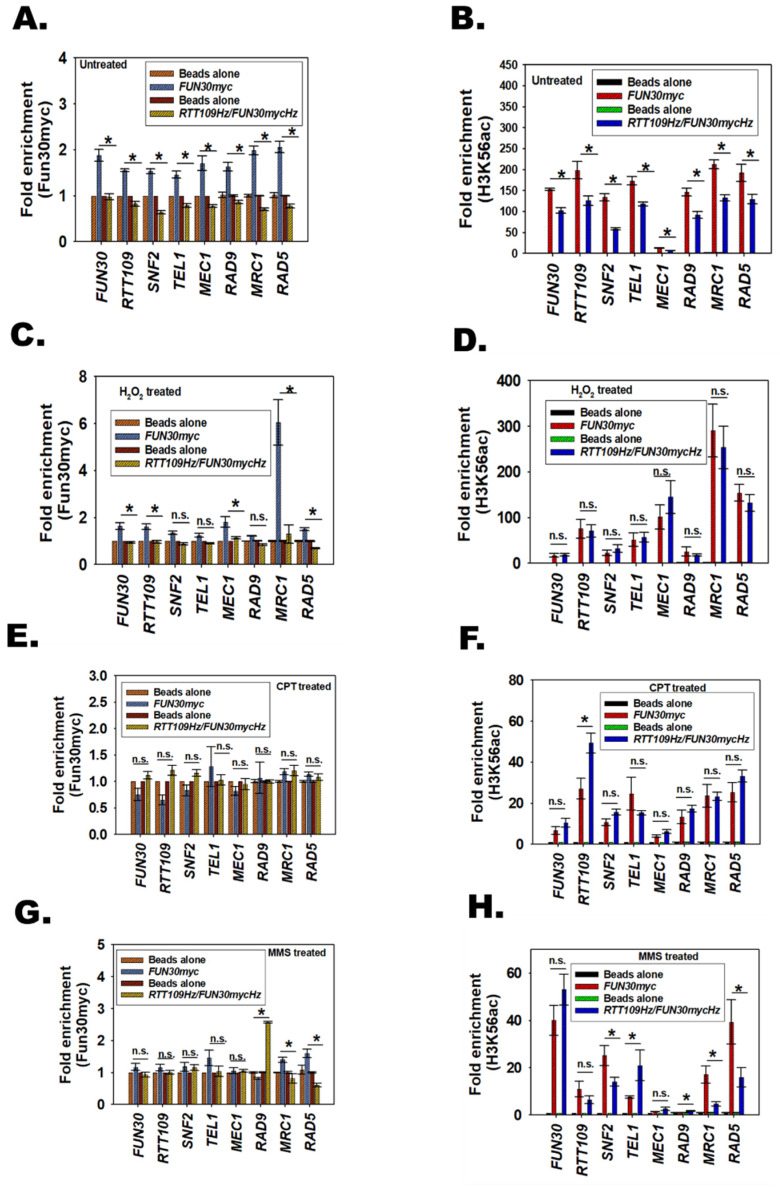
The response of *RTT109Hz*/*FUN30Hz* to DNA damage is regulated by the occupancy of Fun30 and H3K56ac on *RAD9*, *MRC1*, and *RAD5* genes. (**A**). The occupancy of Fun30myc was analyzed on *FUN30*, *RTT109*, *SNF2*, *TEL1*, *MEC1*, *RAD9*, *MRC1*, and *RAD5* promoters in *FUN30myc* and *RTT109Hz*/*FUN30mycHz* in untreated cells. (**B**). The occupancy of H3K56ac was analyzed on *FUN30*, *RTT109*, *SNF2*, *TEL1*, *MEC1*, *RAD9*, *MRC1*, and *RAD5* promoters in *FUN30myc* and *RTT109Hz*/*FUN30mycHz* in untreated cells. (**C**). The occupancy of Fun30myc was analyzed on *FUN30*, *RTT109*, *SNF2*, *TEL1*, *MEC1*, *RAD9*, *MRC1*, and *RAD5* promoters in *FUN30myc* and *RTT109Hz*/*FUN30mycHz* in cells treated with H_2_O_2_ (7.5 mM). (**D**). The occupancy of H3K56ac was analyzed on *FUN30*, *RTT109*, *SNF2*, *TEL1*, *MEC1*, *RAD9*, *MRC1*, and *RAD5* promoters in *FUN30myc* and *RTT109Hz*/*FUN30mycHz* in cells treated with H_2_O_2_ (7.5 mM). (**E**). The occupancy of Fun30myc was analyzed on *FUN30*, *RTT109*, *SNF2*, *TEL1*, *MEC1*, *RAD9*, *MRC1*, and *RAD5* promotes in *FUN30myc* and *RTT109Hz*/*FUN30mycHz* in cells treated with CPT (100 µM). (**F**). The occupancy of H3K56ac was analyzed on *FUN30*, *RTT109*, *SNF2*, *TEL1*, *MEC1*, *RAD9*, *MRC1*, and *RAD5* promoters in *FUN30myc* and *RTT109Hz*/*FUN30mycHz* in cells treated with CPT (100 µM). (**G**). The occupancy of Fun30myc was analyzed on *FUN30*, *RTT109*, *SNF2*, *TEL1*, *MEC1*, *RAD9*, *MRC1*, and *RAD5* promoters in *FUN30myc* and *RTT109Hz*/*FUN30mycHz* in cells treated with MMS (0.02%). (**H**). The occupancy of H3K56ac was analyzed on *FUN30*, *RTT109*, *SNF2*, *TEL1*, *MEC1*, *RAD9*, *MRC1*, and *RAD5* promoters in *FUN30myc* and *RTT109Hz*/*FUN30mycHz* in cells treated with MMS (0.02%). The data is presented as average ± s.e.m. of three biological replicates. * *p* < 0.05.

**Figure 8 jof-08-00559-f008:**
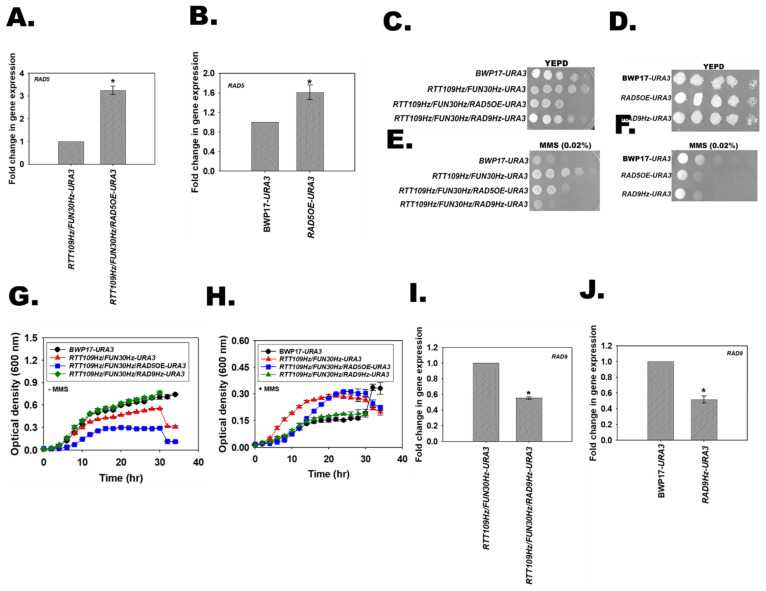
*RAD9* deletion in *RTT109Hz/FUN30Hz* rescues the resistance to MMS. (**A**). Expression of *RAD5* was analyzed in *RTT109Hz/FUN30Hz-URA3* and *RTT109Hz*/*FUN30Hz*/*RAD5OE-URA3* using qPCR. (**B**). Expression of *RAD5* was analyzed in BWP17-*URA3* and *RAD5* overexpression (*RAD5OE-URA3*) strain. (**C**). Growth of BWP17*-URA3*, *RTT109Hz*/*FUN30Hz-URA3*, *RTT109Hz*/*FUN30Hz*/*RAD5OE-URA3*, and *RTT109Hz*/*FUN30Hz*/*RAD9Hz-URA3* on YEPD plate. (**D**). Growth of BWP17-*URA3*, *RAD5OE-URA3*, and *RAD9Hz-URA3* on YEPD plate. (**E**). Sensitivity of BWP17*-URA3*, *RTT109Hz*/*FUN30Hz-URA3*, *RTT109Hz*/*FUN30Hz*/*RAD5OE-URA3*, and *RTT109Hz*/*FUN30Hz*/*RAD9Hz-URA3* to MMS (0.02%) was evaluated using plate assays. (**F**). Sensitivity of BWP17-*URA3*, *RAD5OE-URA3*, and *RAD9Hz-URA3* to MMS (0.02%) was evaluated using plate assays. Growth curve of BWP17*-URA3*, *RTT109Hz*/*FUN30Hz-URA3*, *RTT109Hz*/*FUN30Hz*/*RAD5OE-URA3*, and *RTT109Hz*/*FUN30Hz*/*RAD9Hz-URA3* in the (**G**). absence and (**H**). presence of MMS (0.02%). (**I**). The expression of *RAD5*9 was analyzed in *RTT109Hz*/*FUN30Hz-URA3* and *RTT109Hz*/*FUN30Hz*/*RAD9Hz-URA3* strains using qPCR. (**J**). Expression of *RAD5*9 was analyzed in BWP17*-URA3* and *RAD9Hz-URA3* strains using qPCR. The qPCR data are presented as average ± s.e.m. of three biological replicates. * *p* < 0.05. In the case of plate assays, five-fold serial dilutions were prepared and spotted on respective plates. The plates were incubated at 30 °C and imaged after 24 h.

## Data Availability

Not Applicable.

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
