# Peer review of "Fun30 and Rtt109 Mediate Epigenetic Regulation of the DNA Damage Response Pathway in C. albicans"

_jof, 2022, doi:10.3390/jof8060559_

Round 1

Reviewer 1 Report

All of my concerns have been addressed and I have no further suggestions. I am sorry to hear that your research was impacted by COVID and am looking forward to seeing your future contributions.

Author Response

We would like to thank the reviewer for their kind comments.

Reviewer 2 Report

The article by Maurya et al deals with characterization of C. albicans Fun30 and Rtt109 proteins in regulating DNA damage pathways. The article is an interesting read. However, there are some major concerns that need to be addressed. These are as follows:

1)Line 153: 

Since ATPase activities were analyzed it would be useful to analyze ATP levels in the WT, mutant and the overexpressed FUN30 cells. ATP levels can be measured using renilla luciferase assay following the published article:

https://pubmed.ncbi.nlm.nih.gov/34207384/

Fun30 activity may alter ATP levels in the cells.

2) Section 2.5: Did the authors sequenced once they cloned into the plasmid to verify that no mutations were generated in the transformants? Please clarify.

3)Section 2.9: Was Ca FUN30 codon optimized when it was integrated into Sc?Please clarify.

4)Section 2.11: How were the RNAs quantified? How were the integrity and purity of the RNA assessed? Was DNASEI step performed before cDNA synthesis to remove contaminating RNA? Please clarify

5)Section 2.11: Did the authors used any kits to generate cDNA? Please clarify.

6) Section 2.11: How were the fold changes calculated? Did the authors used Livak’s delta delta Ct method? If so the paper needs to be cited.

7)Section 2.11: Were melting curves performed to determine the specificity of the qPCR? 

8)Section 2.15: How were the doubling times calculated? Please Clarify.

9)Section 2.17: Why MIC50 was used instead of standard MIC80 or MIC90? Please clarify.

10) Through out the paper the authors showed significance data without stating what statistics were used to determine the significance. Please disclose the test name and show the respective p-values for better clarity. p-values are only mentioned in figure 5 and 6 without the mention of the name of the statistical analysis used to generate them.

11)Fig 3: Was qPCR performed in biological replicates? If so the number of replicates need to be mentioned in the figure legend for better clarity.

12)Line 587: "Statistically Significant"- How the statistical significance was determined? Mention the p-values.

Author Response

Reviewer#2 (comments for the author)

1)Line 153: 

Since ATPase activities were analyzed, it would be useful to analyze ATP levels in the WT, mutant and the overexpressed FUN30 cells. ATP levels can be measured using renilla luciferase assay following the published article:

https://pubmed.ncbi.nlm.nih.gov/34207384/

Fun30 activity may alter ATP levels in the cells.

OUR RESPONSE: The ATP-dependent chromatin remodelers are intimately connected to metabolism.  However, none of these proteins have been shown to be involved in ATP production.  Thus, neither S. cerevisiae Fun30 nor human SMARCAD1 have been shown to modulate ATP production.  Nuclear sources of ATP synthesis have lately been investigated in addition to Mitochondrial sources.  The importance of chromatin rearrangement in DNA damage and stress responses, for example, necessitates the existence of a PARP1/PARG/NUDIX5-dependent pathway (Wright RH, Fernandez-Fuentes N, Oliva B, Beato M. Insight into the machinery that oils chromatin dynamics. Nucleus. 2016 Nov;7(6):532-539. doi: 10.1080/19491034.2016.1255392. Epub 2016 Nov 28. PMID: 27893319; PMCID: PMC5214059). However, these pathways have not yet been linked to Fun30 or any of the ATP-dependent chromatin remodelers though it is speculated that these proteins possibly use the nuclear ATP for their function.  It is also believed that the cellular ATP generated by mitochondria remains unaltered.

As our focus was on understanding how Fun30 regulates DNA damage response by regulating the transcription of DNA damage response genes, we did not investigate the levels of ATP in the wild type and mutant cells. We believe this avenue would be interesting to explore in the future.  However, if the reviewer feels that analysis of ATP levels is essential, we can estimate the amount of ATP in wild type, heterozygous and null mutants. We will not be able to use the system recommended by the reviewer as we do not have a luminometer. We can use a colorimetric/fluorescence-based assay.  It will take us 6-8 weeks to complete the experiments as the kit has to be shipped from US/UK and it takes at least 4-6 weeks to reach us.

2) Section 2.5: Did the authors sequenced once they cloned into the plasmid to verify that no mutations were generated in the transformants? Please clarify.

OUR RESPONSE:  We used Phusion Plus DNA polymerase (Catalog # F630S; Thermo Fisher Scientific, USA).  We have clarified the enzyme in the revised manuscript.  As the fidelity of this enzyme is >100X of the Taq polymerase, therefore, we did not sequence any of the clones.  In case of the site-directed mutant, we confirmed the mutation by sequencing.

3)Section 2.9: Was Ca FUN30 codon optimized when it was integrated into Sc?Please clarify.

OUR RESPONSE: No, we have not optimized Ca FUN30 codon when it was integrated into S. cerevisiae.  We were aware of the differences in CUG codon usage between the two organisms and were initially concerned about codon biases.  However, we observed that Candida albicans FUN30 was able to functionally complementation both S. cerevisiae FUN30 and human SMARCAD1 even without codon optimization.  It is quite probable that the leucine amino acid coded by the CUG codon of S. cerevisiae has no effect on the functional conformation of Candida albicans FUN30 protein.

In the revised manuscript, we have stated that we have not codon optimized C. albicans FUN30 before expressing it either in S. cerevisiae or in human HeLa cells.

4)Section 2.11: How were the RNAs quantified? How were the integrity and purity of the RNA assessed? Was DNASEI step performed before cDNA synthesis to remove contaminating RNA? Please clarify

OUR RESPONSE:  RNA samples were quantified using NanoDrop Spectrophotomer (Thermo Fisher Scientific, USA).  The 260/280 as well as the 260/230 ratio was measured using this instrument.  A 260/280 ratio of 2 while 260/230 ratio above 2 indicates pure RNA.  We also do run the RNA samples on formaldehyde agarose gels to check the quality of the isolated sample. To segregate RNA samples and evaluate their integrity on agarose and formaldehyde-agarose (denaturing) gels, we utilized MOPS Buffer as the running buffer for RNA electrophoresis.

We used to perform a DNase1 treatment step before cDNA synthesis to remove contaminating DNA from RNA samples but found that the RNA was degraded as the DNase I digestion buffer can cause (chemically induced) strand scission of RNA when heated.  Therefore, we rely on the Nanodrop measurements to assess the quality of the RNA.  If the ratios are less than 2, then the RNA is discarded.

5)Section 2.11: Did the authors used any kits to generate cDNA? Please clarify.

OUR RESPONSE:  We did not utilize a kit to synthesize the cDNA. We purchased random hexamer primers and dNTPs from Thermo Fisher Scientific, USA.  The reverse transcriptase enzyme and RNase inhibitor were purchased from New England Biolabs, USA.  We have included this in the revised manuscript.

6) Section 2.11: How were the fold changes calculated? Did the authors used Livak’s delta delta Ct method? If so the paper needs to be cited.

OUR RESPONSE: Yes, the delta delta Ct method was used to calculate the relative fold gene expression of samples.  We have clarified this point and included the citation in the revised manuscript.

7)Section 2.11: Were melting curves performed to determine the specificity of the qPCR? 

OUR RESPONSE: Yes, melting curves were performed to determine the specificity of the qPCR. We always make sure that there is a single peak for an amplicon which is typically interpreted as representing a pure, single amplicon. We also perform the standard agarose gel electrophoresis for analyzing the products of qPCR. The presence of a single band indicates a single product.

8)Section 2.15: How were the doubling times calculated? Please Clarify.

OUR RESPONSE:  *The cell growth was measured by recording the absorbance at 600 nm. We plottedln(Absorbance) as a function of time and fitted the data to the exponential equation Y=Ae^BX to fit the data where Y is ln[absorbance], X is time (hr), A represents the initial amount and B represents the growth factor. The doubling time was calculated using the log phase of the curve and equation: Doubling time = ln2/B.

9)Section 2.17: Why MIC50 was used instead of standard MIC80 or MIC90? Please clarify.

OUR RESPONSE:  We agree with the reviewer that it is a standard practice to calculate MIC80 or MIC90.  As we were reporting resistance, we decided to calculate the MIC50.  However, we have also calculated both MIC80 and MIC90. 

The MIC80 was 0.04% ± 0.008% (P = 0.004) for BWP17, 0.04% ± 0.005% (P = 0.004) for RTT109Hz, 0.04%± 0.011% (P = 0.004) for FUN30Hz, and 0.41% ± 0.08% (P = 0.004) for RTT109Hz/FUN30Hz

The MIC90 was 0.05% ± 0.009 (P = 0.004) for BWP17, 0.04 % ± 0.006 (P = 0.004) for RTT109Hz, 0.04% ± 0.013 % (P = 0.004) for FUN30Hz, and 0.46% ±0.09% (P = 0.004) for RTT109Hz/FUN30Hz.

Thus, there is no difference between MIC80 and MIC90 values were not much. Further, as compared to the wild type BWP17 strain, the overall frequency of MMS resistance in the evaluated data set was 10-fold higher in RTT109Hz/FUN30Hz strain.

As suggested by the reviewer, we have replaced the MIC50 with MIC80 values to conform with the standard practice.

10) Throughout the paper the authors showed significance data without stating what statistics were used to determine the significance. Please disclose the test name and show the respective p-values for better clarity. p-values are only mentioned in figure 5 and 6 without the mention of the name of the statistical analysis used to generate them.

OUR RESPONSE: To determine the significance, we used the paired t-test available in Sigma Plot Software. We consider a P-value of < 0.05 to be significant.

In the revised manuscript, we have included the section in the Materials and Methods specifying how the statistical significance was estimated.

Statistical analysis:  All qPCR and ChIP experiments are reported as average ± standard error of mean (SEM) of three independent (biological) experiments unless otherwise specified.  Each independent experiment was performed as at least two technical replicates.  The statistical significance (p-value) was calculated using paired t-test available in Sigma Plot.  The differences were considered significant at p <0.05.

As suggested by the reviewer, we have also included the P values for better clarity.

11)Fig 3: Was qPCR performed in biological replicates? If so the number of replicates need to be mentioned in the figure legend for better clarity.

OUR RESPONSE:  Yes, all qPCR experiments are performed in three biological replicates with two technical replicates performed for each biological replicate.   We have clarified it in the revised manuscript.

12)Line 587: "Statistically Significant"- How the statistical significance was determined? Mention the p-values.

OUR RESPONSE:  The statistical significance was determined using paired t-test in Sigma Plot. We have now mentioned the p-values in the revised manuscript.

Round 2

Reviewer 2 Report

The authors have answered to all my queries satisfactorily and the manuscript has greatly improved.

Section “1.14” immunofluorescence in the materials and methods should be section “2.14”

This manuscript is a resubmission of an earlier submission. The following is a list of the peer review reports and author responses from that submission.

Round 1

Reviewer 1 Report

General Impression

The authors summarize a large body of work aimed at characterizing the function of Fun30, an ATP-dependent chromatin remodeling enzyme, in the epigenetic regulation of DNA repair enzyme expression in Candida albicans. It is concluded that Fun30 and Rtt109-mediated histone acetylation co-regulate expression of DNA repair and damage response genes in response to selected genotoxic agents. The experimental approach is solid, including strain construction, survival studies, immunostaining, western blotting, qPCR and ChIP assays. Experiments are well-described, appropriate controls are included, data analysis thorough and conclusions are supported by evidence. While the article does not fully elucidate the complex relationship of Fun30/Rtt109 interactions and their influence on gene expression outside of the narrow field of DNA repair, it nevertheless makes a valuable contribution to our evolving understanding of epigenetic regulation in Candida and is deserving of publication.

Points of criticism

  1. Functional complementation of human and baker’s yeast mutations by Candida The authors are surely aware of the differential usage of the CUG codon in Candida, and it looks like no effort was made to adapt the cloned CaFun30 gene sequence to human or Saccharomyces codon usage. It is thus somewhat surprising that that Candida sequence functions so well in the other organisms.

Could the authors comment on this?  

  1. Viability of fun30D/fun30D It is concluded that the homozygous fun30 deletion mutant might be inviable since it could not be obtained with a two-step deletion protocol. The argument is presented well. However, the Candida Genome Database rightly or wrongly lists the homozygous null mutant as being “viable”.

Could this be addressed?

  1. Discussion of Fun30/Rtt109 role in H2O2 resistance. The project explores the effect of Fun30/Rtt109 on the somewhat narrow area of DNA repair. While it would probably be prohibitive to broaden the scope of the project to address epigenetic effects on a genome-wide scale, the focus on DNA repair phenotypes is limiting the discussion of the H2O2-sensitivity phenotype. Oxidative stress is the most severe environmental threat to albicans inside the host, and the fungus has powerful antioxidant defense mechanism. The reader is left to wonder about the effect of H3K56 acetylation and Fun30 on the expression of other radical defense and repair mechanisms during H2O2 stress (e.g. superoxide dismutase).

A discussion of this limitation would be in order.

  1. More tables. Some figures present an overwhelming amount of data (e.g. Fig. 6). This reviewer had to draw a table to facilitate understanding, and it is likely that the readers would benefit from such tables as well.

Can a summary table be added to the more complicated figures 6 and 7?

  1. Fewer tables. Figure 8 G,H and table 1 show essentially the same data. Consider cutting out one of the two.

Writing and grammar

There are dozens of grammatical mistakes that warrant attention. Most could probably be found with a simple grammar check (subject/verb agreement, missing words).

Author Response

Reviewer’s concern-

  1. Functional complementation of human and baker’s yeast mutations by The authors are surely aware of the differential usage of the CUG codon in Candida, and it looks like no effort was made to adapt the cloned CaFun30 gene sequence to human or Saccharomyces codon usage. It is thus somewhat surprising that the Candida sequence functions so well in the other organisms.

Our response-

Yes, we are aware of the differential usage of the CUG codon in Candida albicans and initially were worried about the codon biases.  The complete functional complementation of Candida albicans Fun30 in the human or S. cerevisiae system has been a surprise to us too for we thought we will see only partial complementation.  As there was complete functional complementation, we have not pursued adapting the codon of the cloned CaFun30 gene sequence to human or Saccharomyces codon usage.  The CUG codon in C. albicans encodes for serine while it encodes for leucine in S. cerevisiae.  We speculate that as both of these are small non-charged amino acids, the function and the activity of the protein expressed in the heterologous system may not be affected. 

Reviewer’s concern-

  1. Viability of fun30D/fun30D It is concluded that the homozygous fun30 deletion mutant might be inviable since it could not be obtained with a two-step deletion protocol. The argument is presented well. However, the Candida Genome Database rightly or wrongly lists the homozygous null mutant as being “viable”.

Our response-

Yes, the Candida Genome Database states that the homozygous null mutant is ‘viable’.  It is also not essential in S. cerevisiae. We also observed that when both copies of RTT109 are deleted, the Fun30 expression could not be observed by Western blot.  This was one of the reasons that we made multiple attempts to obtain the null mutant.  In the discussion, we have left the question of Fun30 being essential open.  We do not know if, in the future, we might be able to make null homozygous mutant or not.  One possibility is to use Tet-OFF/Tet-ON inducible vectors and see whether we can prove the essentiality of the gene.  We could not do this experiment at this point because we could not obtain this vector due to the lockdown necessitated by the Covid pandemic.

Reviewer’s concern-

  1. Discussion of Fun30/Rtt109 role in H2O2 The project explores the effect of Fun30/Rtt109 on the somewhat narrow area of DNA repair. While it would probably be prohibitive to broaden the scope of the project to address epigenetic effects on a genome-wide scale, the focus on DNA repair phenotypes is limiting the discussion of the H2O2-sensitivity phenotype. Oxidative stress is the most severe environmental threat to albicans inside the host, and the fungus has powerful antioxidant defense mechanism. The reader is left to wonder about the effect of H3K56 acetylation and Fun30 on the expression of other radical defense and repair mechanisms during H2O2 stress (e.g. superoxide dismutase).

Our response-

In our study, we were more concerned about the DNA damage response pathways. So, we have only measured the expression of some of the DNA damage response genes like initiator kinases TEL1 and MEC1 and the genes involved in DNA damage checkpoint pathways like RAD9 and MRC1.  Within this narrow range, we found that only Fun30 was playing a role in regulating the expression of these genes in the presence of oxidative stress.  We agree that it limits the discussion on the role of Fun30 and Rtt109 in oxidative stress.  Further, in the absence of reports discussing the role of these two proteins in oxidative stress, we could not expand the discussion section.  We have planned for studies to compare Fun30 and H3K56ac occupancy at a genomic-wide scale using ChIP-seq in the presence of DNA damage caused by oxidative stress and MMS.

Reviewer’s concern-

  1. More tables. Some figures present an overwhelming amount of data (e.g. Fig. 6). This reviewer had to draw a table to facilitate understanding, and it is likely that the readers would benefit from such tables as well. Can a summary table be added to the more complicated figures 6and 7?

Our response-

In the revised manuscript, we have presented Summary Tables for the data presented in Figures 5A, 6, and 7.  The Summary Tables are included in the Supplementary information as Supplementary Table S6, S7, and S8 respectively.

Reviewer’s concern-

  1. Fewer tables. Figure 8 G, H, and table 1 show essentially the same data. Consider cutting out one of the two.

Our response-

We have now shifted Table 1 into Supplementary Information as Supplementary Table 9.  As this table showed the duplication time, we thought it was necessary to be included in the manuscript.

Reviewer’s concern-

Writing and grammar-There are dozens of grammatical mistakes that warrant attention. Most could probably be found with a simple grammar check (subject/verb agreement, missing words).

Our response-

We apologize for the errors.  We have re-read the manuscript and have tried to correct all the mistakes.

Reviewer 2 Report

See attached file

Author Response

Reviewer’s concern-

  1. One of the major issues with the manuscript was the formatting, which made it difficult to read. The figure legends were not always aligned with the figures and part of the text was incorporated into the main body versus the figure legends. Additionally, most of the figures were too small, making it hard to assess the data. This was particularly true for Figures 1, 3, 4, 8, and S1A. Finally, in the supplemental data, Figure S4 was overlaid on S3 so this data could not be assessed. I realize that this may not be entirely the author's fault but these issues need correcting.

Our response-

We have ensured that Figure S3 and Figure S4 are visible in the revised manuscript.  We have also tried to address the formatting issue.

Reviewer’s concern-

  1. There are several important controls and analyses that are missing from the study. First, the experiment showing that Fun30 can rescue DNA damage signaling in HeLa cells after SMARCAD1 knockdown is an excellent experiment to show conservation of function across species (Fig 1 B-D). However, the authors did not show a Western blot of SMARCAD1 knockdown or expression of Fun30 in the HeLa cells. These controls are essential to validate their findings. Second, in Fig 3E and 4D, the authors show decreased levels of Fun30 but it is hard to determine from the blot how much the levels decrease. At minimum, quantification of the blots shown should be included. Even better would be a graph showing multiple trials. Third, in Fig S4A-B, wild-type controls are not shown for comparison. This is especially important since there is an additional band in the gel that is not accounted for.

Our response-

We have now included the western blot of SMARCAD1 knockdown in the HeLa cells (Supplementary Fig. S2H and I).  CaFun30 overexpression in siSMARCAD1 cell was also checked by PCR amplification.  The gel picture of the same is included in the Supplementary file (Supplementary Fig. S2J).

Quantification of the blots shown in Fig. 3E and Fig. 4D is now shown in the revised manuscript as Supplementary Fig. 3G and H respectively.

Yes, in Figures S4A and B, we observed a non-specific band along with strong bands of 1.4 kb and 2.4 kb corresponding to the specific size of HIS1 and ARG4 markers.  We concur with the reviewer that we should have included wild-type controls in the gels.  We apologize for this oversight.  Unfortunately, our laboratories are closed because of extended lockdown due to the Covid pandemic and we could not revisit these gels.

We, however, did confirm both these mutants by qPCR where we found reduced expression of RTT109 (Figure 4A), and by western blot where we found H3K56ac (Figure 4C) was also reduced.

Reviewer’s concern-

  1. A key finding of this study is that Fun30 is conserved in albicans with both Fun30 in S. cerevisiae and SMARCAD1 in humans. The authors present phylogenetic analysis in Fig S1A. However, some additional information is needed to fully understand the extent of this conservation. This includes the percent sequence conservation between C. albicans, S. cerevisiae, and humans both across the entire protein and in the RecA-like and CUE domains. Also regarding this point, the discussion also does not have any references to previous work on SMARCAD1 or much on S. cerevisiae Fun30 to put their results in context of previous findings. Additional discussion of these points would greatly improve the manuscript.

Our response-

We have added information about the percent sequence conservation across the entire protein, RecA-like domains, and CUE domain in section 3.1.  We have also included these points in the discussion.

Reviewer’s concern-

  1. The findings that Fun30/Rtt109 heterozygotes lead to sensitivity to H2O2, no effect with CPT, and resistance to MMS treatments was quite perplexing and authors did not address whether there is previous literature to support these differences or what might be happening. This point needs to be discussed or clarified in the results and/or discussion. Are the changes in the DDR factors enough to account for the differences observed? Did you test for direct changes in the DDR signaling? Why would these changes in gene expression be dependent on the type of DNA damage? Other potential explanations?

Our response-

In Candida albicans, it is already reported that Rtt109  is involved in DNA damage response.  Deletion of Rtt109 renders the fungal cells hypersensitive to oxidative stress, MMS, and CPT.  In our laboratory too, we found that Rtt109 -/- cells were hypersensitive to oxidative stress.  The cross-talk between Rtt109 and Fun30 has not been investigated, which was the focus of our studies.

It has been already reported that during DNA damage, two different checkpoints are activated to protect the cell.  First is the DNA damage checkpoint (DDC), which is mediated by Rad9, and second is the DNA replication checkpoint (DRC), which is mediated by Mrc1. Rad9 is known to activate in all the phases of the cell cycle but Mrc1 is only functional during S-phase. It is also reported that Mrc1 depleted cell functions normally due to the presence of Rad9 (Where Rad9 takes over the function of Mrc1 during the S-phase).  Regardless of the pathway, the ultimate goal is the phosphorylation of Rad53, which activates DNA repair genes in the cell.

On the other hand, Rad5, an E3-ubiquitinase, is required for the post-replication bypass at the stalled replication fork. When Rad5 accumulation is low in the stalled replication fork, it activates the translesion synthesis, and when it accumulates more, an error-free lesion bypass pathway takes over.

In our study, RTTT109Hz/FUN30Hz strain showed differential phenotypes in the presence of three different genotoxic reagents, which we believe can be understood by these following points-

  1. In the presence of H2O2, we observed the downregulation of TEL1, MEC1, RAD5, RAD9, and MRC1 in the RTTT109Hz/FUN30Hz Because of decreased expression of these initiator kinases and mediator proteins, both DDC and DRC may not be activated properly. So, ultimately Rad53 phosphorylation and therefore, activation of DNA damage repair is impaired. Thus, the cell will not be able to mount a response to the DNA damage caused by oxidative stress leading to a sensitive phenotype.
  2. In the presence of CPT, the transcript levels of MEC1, RAD9, and MRC1 were not altered in RTTT109Hz/FUN30Hz Therefore, the DNA damage response pathway is operational in the mutant strain. Consequently, the cells were able to mount a response to the DNA damage resulting in normal growth in the mutant strain as compared to the wild type.
  • In the presence of MMS, the expression of MEC1 was unchanged indicating the DNA damage response was probably functioning. However, the expression of MRC1 and RAD5 was downregulated while the expression of RAD9 was upregulated in the RTT109Hz/FUN30Hz The overexpression of RAD9 could enhance the repair efficiency in the mutant strain as compared to the wild-type strain. The decreased expression of RAD5 might result in a bypass of the replication fork by translesion synthesis. Therefore, more growth in the mutant strain is observed as compared to the wild-type strain, exhibiting resistance phenotype.

One of the major problems in working with C. albicans is the lack of availability of antibodies.   Therefore, we were unable to check the direct occupancy of Rad5, Rad9, and Mrc1 at the DNA damage site. However, we agree with the reviewer that such experiments need to be done.  One way of doing these experiments is to tag the endogenous gene with myc or His and then use antibodies against these moieties.  We have planned these experiments for the future.  

As we know, different kinds of DNA damage may activate different signaling mechanisms leading to the formation of specific protein complexes required for that particular DNA damage response pathway.  Therefore, one of the ways how Fun30 and Rtt109 could be differentially regulating the DNA damage response genes is via the formation of specific protein complexes.  Therefore, the next step is to understand the composition of these protein complexes and we have planned for such studies in the future.  

Reviewer’s concern-

  1. The discussion was mostly a restatement of the results. As mentioned above, putting the results in context of the other studies of Fun30 homologues and what is known about epigenetic regulation of DDR factors is needed.

Our response-

We have now included discussion points on epigenetic regulation of DDR factors as well as other studies of Fun30 homologs in the revised manuscript.

Reviewer’s concern-

  1. For clarity, adding that JKM179 is an cerevisiae strain would be help those not familiar with the strain. It was confusing why you were able to create a full KO in these cells but not in the next section without that information.

Our response-

Thank you for the suggestion.  We have now clarified that JKM179 is an S. cerevisiae strain in the revised manuscript.

Reviewer 3 Report

The major finding of this study is the discovery of the functional homology of S. cerevisiae Fun30 in C. albicans. The authors go on to show that like S. cerevisiae Fun30 regulates the DNA damage response by regulating the expression of several DDR proteins, particularly Tel1 and Mec1. Furthermore, they find that Fun30 is regulated by histone acetyltransferase Rtt109. Interestingly, in C. albicans Fun30 heterozygotes, DDR response genes are differential regulated depending on the type of DNA damage. Overall, the study was well designed with a clear hypothesis, methods were well described, and the manuscript well written. This study adds insight into the epigenetic regulation of the DDR in yeast and conserved functions of Fun30. However, there were several concerns, outlined below, that should be addressed prior to publication, particularly the formatting of the manuscript and addition of several key controls. If these concerns can be adequately addressed, I would recommend publication of this study in your journal.

  1. One of the major issues with the manuscript was the formatting, which made it difficult to read. The figure legends were not always aligned with the figures and part of the text was incorporated into the main body versus the figure legends. Additionally, most the figures were too small, making it hard to assess the data. This was particularly true for Figures 1, 3, 4, 8 and S1A. Finally, in the supplemental data, Figure S4 was overlaid on S3 so this data could not be assessed. I realize that this may not be entirely the authors fault but these issues need correcting.

  1. There are several important controls and analyses that are missing from the study. First, the experiment showing that Fun30 can rescue DNA damage signaling in HeLa cells after SMARCAD1 knockdown is an excellent experiment to show conservation of function across species (Fig 1 B-D). However, the authors did not show a Western blot of SMARCAD1 knockdown or expression of Fun30 in the HeLa cells. These controls are essential to validate their findings. Second, in Fig 3E and 4D, the authors show decreased levels of Fun30 but it is hard to determine from the blot how much the levels decrease. At minimum, quantification of the blots shown should be included. Even better would be a graph showing multiple trials. Third, in Fig S4A-B, wild-type controls are not shown for comparison. This is especially important since there is an additional band in the gel that is not accounted for.

  1. A key finding of this study is that Fun30 is conserved in C. albicans with both Fun30 in S. cerevisiae and SMARCAD1 in humans. The authors present phylogenetic analysis in Fig S1A. However, some additional information is needed to fully understand the extent of this conservation. This includes the percent sequence conservation between C. albicans, S. cerevisiae and humans both across the entire protein and in the RecA-like and CUE domains. Also regarding this point, the discussion also does not have any references to previous work on SMARCAD1 or much on S. cerevisiae Fun30 to put their results in context of previous findings. Additional discussion of these points would greatly improve the manuscript.

  1. The findings that Fun30/Rtt109 heterozygotes leads to sensitivity to H2O2, no effect with CPT and resistance to MMS treatments was quite perplexing and authors did not address whether there is previous literature to support these differences or what might be happening. This point needs to be discussed or clarified in the results and/or discussion. Are the changes in the DDR factors enough to account for the differences observed? Did you test for direct changes in the DDR signaling? Why would these changes in gene expression be dependent on the type of DNA damage? Other potential explanations?

  1. The discussion was mostly a restatement of the results. As mentioned above, putting the results in context of the other studies of Fun30 homologues and what is known about epigenetic regulation of DDR factors is needed.

  1. For clarity, adding that JKM179 is a S. cerevisiae strain would be help those not familiar with the strain. It was confusing why you were able to create a full KO in these cells but not in the next section without that information.

Author Response

Reviewer’s concern-

  1. The authors were unable to create a homozygous null fun30 mutant, which does not mean that Fun30 is an essential gene. A simple and effective method of addressing this issue is to delete both Fun30 alleles from cells harbouring a URA3-FUN30 plasmid, and then test if the cell can lose the plasmid.

Our response-

We did think about doing the suggested experiment but episomal plasmids are not available for C. albicans as the organism ejects them out.  Therefore, it is not possible to do this experiment.  This is the reason why we integrated a third copy of the gene and showed that the two original copies of the gene can be deleted.  Of course, this does not answer whether Fun30 is essential conclusively.  We have been thinking of other options.  For example, using a Tet-OFF or Tet-ON inducible systems for creating conditional mutants.  We will be doing these experiments once the extended lockdown mandated by the Covid pandemic is lifted.

Reviewer’s concern-

  1. The authors found no effect of FUN30Hz on genotoxin resistance and expression of a group of DNA damage response genes, which suggest that “… one copy of FUN30 is sufficient to drive the response to genotoxic stress.” or “… the changes in the gene expression are not relevant (to genotoxin resistance).”  As a result, it was problematic to use FUN30Hz to examine Fun30 function, which made the results from experiments involving FUN30Hz intrinsically difficult to interpret.

Our response-

There was indeed no phenotypic effect of these genotoxic reagents in FUN30Hz.  We concur with the reviewer that the role of Fun30 in C. albicans can be studied only if we are successful in creating either a homozygous null or a conditional null mutant. 

As it was not possible to decipher the role of Fun30 using FUN30Hz, we decide to shift our focus to understand the cross-talk between Fun30 and Rtt109 by creating RTT109Hz/FUN30Hz mutant.  At this stage, from the experimental results presented in this paper, we can only say that both Rtt109 and Fun30 are needed for the regulation of DNA damage response genes; however, to understand the contribution of Fun30 only, we will need to look at null mutants.

Reviewer’s concern-

  1. The term co-regulation was ill-defined. The functional relationship between Fun30 and Rtt109 in regulating genes involved in DNA damage response was not re revealed by results in this report.

Our response-

We used the term co-regulation because Rtt109 binds to the FUN30 promoter and regulates its expression.  Similarly, Fun30 binds to the RTT109 promoter and regulates its expression.  However, we have removed the term “co-regulation” in the revised manuscript wherever possible.

Our studies show that Fun30 and Rtt109 regulate the expression of the DNA damage response genes.  We do agree with the reviewer further studies will need to be done to understand the relationship between these two proteins. For example, is Fun30 recruited to the promoters by binding to H3K56ac?  Studies are underway in our laboratory to identify the protein complex formed during genotoxic stress.

Reviewer’s concern-

  1. Several key experiments on the functional relationship of Fun30 and Rtt109 in gene regulation failed to include necessary (single mutant) strains. For example, in Fig. 5 A, B and C, single mutants FUN30mycHz and RTT109Hz were not tested. As such, the epistatic relationship between FUN30 and RTT109 were not revealed. Results shown in Fig. 6 and supplementary Fig. 5 suffered from the same problem.

Our response-

We apologize for not including the data for the single mutant strains in Fig. 5.  The qPCR data for single mutants, in the absence of genotoxic stress, was shown in Figure 3 (for FUN30Hz) and Figure 4 (for RTT109Hz).  Therefore, we did not include the data in this figure.  We can do so if the reviewer wants it for comparison purposes. However, it will be the same as shown in Figures 3 and 4.

In Fig. 5D and E, the spot assays results for the single mutants have been shown along with the double mutants.  We have shown qPCR data for Fun30Hz in the presence of genotoxic stress in Supplementary Figure 3.  We did not observe any change in gene expression as compared to the wild-type strain.  Therefore, we did not do ChIP experiments in this case.  Similarly, as there was no phenotypic change in the RTT109Hz mutant in the presence of genotoxic stress, we did not do qPCR or ChIP.   Instead, we decided to compare the double mutant with the wild-type strain.

Reviewer’s concern-

  1. A major issue of the work of as a whole is that experiments on Fun30 and/or Rtt109 were done in different genetic contexts that did not always yield consistent results.

For example: 

  1. In the presence of MMS, TEL1 is upregulated in RTT109Hz/FUN30mycHz (Fig. 6D), but unchanged or downregulated in RTT109Hz/FUN30Hz (Fig. S5D)
  2. In the presence of CPT, TEL1 is unchanged in RTT109Hz/FUN30mycHz (Fig. 6D), but upregulated in RTT109Hz/FUN30Hz (Fig. S5D).
  3. In the presence of MMS, SNF2 is upregulated in RTT109Hz/FUN30mycHz (Fig. 6D), but downregulated in RTT109Hz/FUN30Hz (Fig. S5D)
  4. In the presence of CPT, SNF2 is unchanged in RTT109Hz/FUN30mycHz (Fig. 6D), but downregulated in RTT109Hz/FUN30Hz (Fig. S5D).

The authors failed to point out thein consistencies or offer any explanation.

Our response-

We concur that we have observed variable expression for some of the genes.  We think that these could be strain-specific differences and further studies are needed to understand why this is happening.  We have now pointed out the inconsistencies in the revised manuscript, as suggested by the reviewer.

Reviewer’s concern-

  1. Panels D, E, F of Supplementary Fig. 3 were covered by the image of Supplementary Fig. 4, making it impossible to evaluate the relevant data.

Our response-

We apologize for the oversight.  We have rectified the error and ensured that all the figures are correctly visible.

Reviewer’s concern-

  1. In section 3.13, the authors’ hypothesis that the degree of genotoxin resistance of the RTT109Hz/FUN30Hz is dependent on the expression level of a select group of DNA damage response factors is over simplified because DNA damage response is an intricate network consisting of many other factors in addition to the group of factors examined.

Our response-

We agree with the reviewer.    In this paper, we have selected only a narrow range of DNA damage response genes and have sought to understand their regulation by Fun30 and Rtt109.  Genome-wide studies need to be done to unravel the entire network.  This was beyond the scope of this paper but we are planning to do it in the future.

Reviewer’s concern-

Minor points:

  1. Rad5 is not a bona fide chromatin remodeling factor

Our response

Rad5 is a DNA-dependent ATPase belonging to the SWI/SNF family [1,2].  It can re-anneal strands like SMARCAL1 [3].  It also possesses ubiquitination activity. 

References:

  1. Johnson, R.E.; Prakash, S.; Prakash, L. Yeast DNA Repair Protein RAD5 That Promotes Instability of Simple Repetitive Sequences Is a DNA-Dependent ATPase. J Biol Chem 1994, 269, 28259–28262.
  2. Gangavarapu, V.; Haracska, L.; Unk, I.; Johnson, R.E.; Prakash, S.; Prakash, L. Mms2-Ubc13-Dependent and -Independent Roles of Rad5 Ubiquitin Ligase in Postreplication Repair and Translesion DNA Synthesis in Saccharomyces Cerevisiae. Mol Cell Biol 2006, 26, 7783–7790, doi:10.1128/MCB.01260-06.
  3. Blastyák, A.; Pintér, L.; Unk, I.; Prakash, L.; Prakash, S.; Haracska, L. Yeast Rad5 Protein Required for Postreplication Repair Has a DNA Helicase Activity Specific for Replication Fork Regression. Molecular Cell 2007, 28, 167–175, doi:10.1016/j.molcel.2007.07.030.

  1. “orf19.6291 encodes for Fun30” is not grammatically correct. It should be “orf19.6291 encodes Fun30”. This mistakes occurs in numerous places.

Our response:

We apologize for the mistake.  We have rectified it in the revised manuscript.

  1. Page 2, line 7, “yeast and fungi” is awkward as yeast is a fungus.

Our response:

We apologize for the mistake.  We have rectified it in the revised manuscript.

  1. Page 9, section 3.5, combine the first 3 paragraphs into one.

Our response-

We have merged the 3 paragraphs in Section 3.5 into one paragraph as suggested by the reviewer.

Round 2

Reviewer 3 Report

There are still some formatting issues, including some parts of the figure legends being in the main text and figure sizes not being increased much, that should be addressed but I assume this can be resolved in the production stages. Besides that, the authors have adequately addressed my concerns and I recommend the paper for publication.